# Multi-Expert Distributionally Robust Optimization for Out-of-Distribution Generalization

**Jinyong Jeong**[1]    **Hyungu Kahng**[2]*    **Seoung Bum Kim**[1]*

[1]Department of Industrial and Management Engineering, Korea University, Seoul
[2]Department of Convergence Business, Korea University, Sejong
{jy_jeong, hgkahng, sbkim1}@korea.ac.kr

## Abstract

Distribution shifts between training and test data undermine the reliability of deep neural networks, challenging real-world applications across domains and subpopulations. While distributionally robust optimization (DRO) methods like GroupDRO aim to improve robustness by optimizing worst-case performance over predefined groups, their use of a single global classifier can be restrictive when facing substantial inter-environment variability. We propose Multi-Expert Distributionally Robust Optimization (MEDRO), a novel extension of GroupDRO designed to address such complex shifts. MEDRO employs a shared feature extractor with $m$ environment-specific expert classifier heads, and introduces a min-max objective over all $m^2$ expert-environment pairings, explicitly modeling cross-environment risks. This expanded uncertainty set captures fine-grained distributional variations that a single classifier might overlook. Empirical evaluations on a range of standard distribution shift benchmarks demonstrate that MEDRO often achieves robust predictive performance compared to existing methods. Furthermore, MEDRO offers practical inference strategies, such as ensembling or gating mechanisms, for typical scenarios where environment labels are unavailable at test time. Our findings suggest MEDRO as a promising step toward resilient and generalizable machine learning under real-world distribution shifts.

## 1 Introduction

Deep neural networks have achieved remarkable success under the assumption that training and test data are drawn from the same distribution. In real-world applications, however, this assumption often breaks down, and even minor deviations—known as *distribution shifts*—can significantly impair performance [1, 2]. Such shifts are common across domains like medical diagnostics (e.g., changing demographics or imaging protocols) and natural language processing (e.g., text from emerging sources), posing serious challenges for model reliability [3, 4].

Two common types of distribution shift are *subpopulation shift* and *domain shift* [5]. Subpopulation shift arises when specific subgroups—such as minority demographics—exhibit different feature or label distributions from the majority population [6, 7]. Domain shift, by contrast, involves broader changes in data-generating processes, such as differences in visual style or linguistic domain [8]. We follow Koh et al. [5] in defining subpopulation shifts as variations within a domain, and domain shifts as those occurring between distinct domains. Both types can degrade performance and require robust generalization approaches [5, 6, 9].

A leading approach is *distributionally robust optimization* (DRO), which minimizes the worst-case risk over a set of possible distributions [10, 6]. In GroupDRO, a prominent DRO variant, the model is

---

*Corresponding authors

trained to minimize the maximum loss across predefined environments, promoting robustness under subpopulation shifts. However, GroupDRO relies on a single global classifier and may struggle when optimal decision boundaries vary substantially between environments [11]. Moreover, its uncertainty set—defined over convex mixtures of environments—may overlook nuanced interactions between them.

To address these limitations, we propose *Multi-Expert* DRO (MEDRO), which extends GroupDRO by assigning a specialized expert head to each environment while sharing a common feature extractor. This design enlarges the uncertainty set from an $(m-1)$-dimensional simplex to an $(m^2-1)$-dimensional one, accounting for all expert–environment combinations. Each expert minimizes risk on its designated environment while also learning to generalize across mismatched inputs from other environments.

We evaluate MEDRO across a variety of distribution shift settings. On subpopulation shift benchmarks [6, 12], MEDRO significantly improves worst-group accuracy. It also generalizes effectively to domain and mixed-shift benchmarks [5], consistently outperforming single-head DRO baselines.

Since environment labels are unavailable at test time, we introduce two inference strategies: a simple ensemble that averages expert outputs, and a gating network that adaptively weighs them based on the input. These strategies enable effective deployment of MEDRO without requiring environment annotations at inference time.

Overall, MEDRO extends the DRO framework by explicitly modeling expert–environment interactions, offering a unified and principled approach to both subpopulation and domain-level shifts.

The remainder of the paper is organized as follows. Section 2 reviews out-of-distribution generalization and DRO. Section 3 provides the requisite background and introduces the proposed MEDRO framework. Section 4 reports experimental findings, and Section 5 concludes with potential future research avenues.

Summary of contributions:

- **Multi-expert DRO formulation:** We propose a principled extension of DRO that assigns a specialized expert head to each environment and optimizes over all expert–environment pairs. This formulation expands the DRO uncertainty set and captures cross-environment variations beyond the capacity of single-head models.

- **Theoretical analysis:** We show that MEDRO extends GroupDRO by explicitly modeling expert–environment interactions beyond group-wise risks. Our formulation recovers GroupDRO as a special case and enjoys convergence guarantees under standard assumptions.

- **Unified approach to subpopulation and domain shift:** By modeling all expert–environment interactions, our approach unifies subpopulation robustness and domain generalization within a single framework, supported by theoretical analysis and empirical results for both types of shifts.

## 2 Related work

### 2.1 Out-of-distribution generalization

Out-of-distribution (OOD) generalization aims to ensure robust model performance when test distributions deviate from those seen during training. A variety of methods address this challenge. One line of work focuses on *invariant feature learning*, which encourages representations that remain predictive across environments [13, 14], often through distribution alignment or adversarial training [15, 16]. Another direction builds on *causal inference*, assuming structural knowledge such as causal graphs or conditional independencies to identify stable predictors [17], and includes approaches that model domain shift as selection bias [18].

Complementary perspectives include *meta-learning*, which optimizes models for cross-domain generalization by simulating distribution shifts during episodic training [19], and *data augmentation*, which enriches the training distribution via transformations or perturbations [20, 21]. Recent work also explores generating synthetic domains [22] or aligning optimization dynamics [23] to further improve generalization.

Despite their diversity, existing methods often rely on fixed assumptions about invariance or training distribution coverage, without explicitly addressing worst-case scenarios. As a result, performance may degrade under severe or unanticipated shifts. This motivates the use of more principled approaches such as distributionally robust optimization (DRO), which explicitly targets worst-case robustness.

## 2.2 Distributionally robust optimization and generalization of robust models

DRO addresses distribution shift by optimizing worst-case risk over an uncertainty set [10, 24, 25]. Classical DRO formulations define an uncertainty set $\mathcal{Q}$ around the empirical training distribution $\widehat{\rho}$, typically using a divergence-based radius (e.g., an $f$-divergence or Wasserstein ball) [24, 25, 26, 27]. Alternatively, formulations based on maximum mean discrepancy define an uncertainty set and establish connections between DRO and regularization in kernel methods [28]. While these approaches offer robustness to localized perturbations in $(\mathbf{x}, y)$, they are often less effective for structured shifts, such as those involving subpopulations or domains.

To address this, early work explored DRO objectives under structured scenarios like label or data-source shift [29, 30]. Building on this direction, GroupDRO [6] defines the uncertainty set over predefined groups, enabling reweighting within observed groups while disallowing shifts beyond their convex hull. By adversarially adjusting group weights, GroupDRO mitigates the risk of underrepresented groups being overlooked, thereby improving worst-group performance on minority groups. Recent work extends GroupDRO to affine combinations of training risks, encouraging robustness beyond the convex hull [9].

However, GroupDRO and its extensions typically rely on a single classifier to represent all groups. While shared models ensure coverage, they may overlook critical inter-group or domain-specific variation [5]. This motivates DRO formulations that can explicitly model interactions between experts and environments—a direction we pursue in this work.

# 3 Methodology

## 3.1 Problem setup

We consider a supervised learning problem in which training data is drawn from $m$ distinct environments (also referred to as groups or domains), indexed by $e = 1, 2, \ldots, m$. Each environment $e$ is characterized by a probability distribution $\mathcal{P}_e$ over the input-output space $\mathcal{X} \times \mathcal{Y}$. We denote a sample from environment $e$ by $(\mathbf{x}, y) \sim \mathcal{P}_e$. The model parameters are denoted by $\theta \in \Theta$, where $\Theta$ is the parameter space, and a loss function $\ell : \mathcal{Y} \times \mathcal{Y} \to \mathbb{R}_+$ quantifies prediction error.

Empirical risk minimization (ERM) is a common strategy that minimizes the average loss across observed environments [31]. However, ERM often struggles when the test distribution deviates from what was seen during training, leading to poor generalization [5, 12].

## 3.2 From ERM to DRO

To tackle distributional shifts, distributionally robust optimization (DRO) focuses on minimizing the worst-case expected loss over an uncertainty set $\mathcal{Q}$. Formally, DRO solves:

$$\min_{\theta \in \Theta} \max_{q \in \mathcal{Q}} \ \mathbb{E}_{(\mathbf{x}, y) \sim q} \big[ \ell \big( f_\theta(\mathbf{x}), y \big) \big]. \tag{1}$$

The design of $\mathcal{Q}$ governs the types of distribution shifts the model becomes robust against. For instance, $\mathcal{Q}$ might comprise all distributions within a specified distance (e.g., a Wasserstein ball) around an empirical measure [24, 25], or consist of mixtures of known group distributions [6].

## 3.3 GroupDRO framework

GroupDRO [6], a well-known form of DRO for multiple environments, defines $\mathcal{Q}$ as the set of convex combinations of $\mathcal{P}_1, \ldots, \mathcal{P}_m$:

$$\mathcal{Q} := \Big\{ \sum_{e=1}^{m} \lambda_e \mathcal{P}_e \ \Big| \ \boldsymbol{\lambda} \in \Delta_m \Big\}, \tag{2}$$

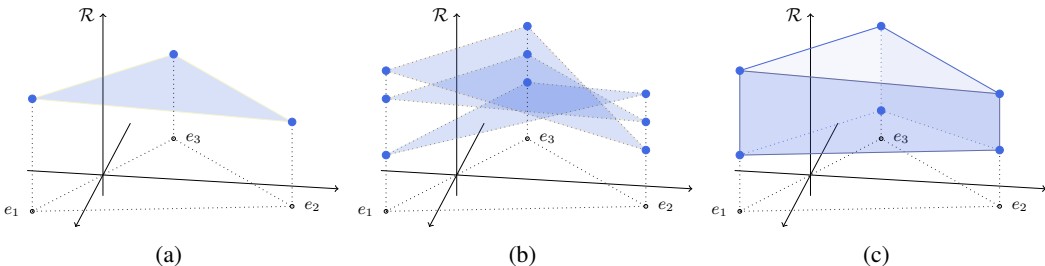

$$
\begin{array}{ccc}
\text{(a)} & \text{(b)} & \text{(c)}
\end{array}
$$

Figure 1: Conceptual visualization of uncertainty sets for $m = 3$ environments. Environments $\mathcal{P}_j$ are represented by distinct locations in the $(x,y)$-plane; the $z$-axis denotes risk. (a) **GroupDRO**: The risk surface (shaded triangle) for a single classifier, defined over the convex hull of the $m = 3$ base environment distributions. GroupDRO optimizes for the worst-case risk on this surface. (b) **MEDRO (Individual Experts)**: Illustrates the three distinct risk surfaces (triangular patches), one for each of MEDRO's $m = 3$ experts ($\omega_i \circ \phi$). Each surface shows how an individual expert $i$ performs across all environments $j$ (i.e., visualizing its $\mathcal{R}_{i,j}$ profile). (c) **MEDRO (Expanded Uncertainty Set)**: The convex hull of all $m^2 = 9$ cross-environment risks ($\mathcal{R}_{i,j}$ points). MEDRO's worst-case component is maximized over this significantly more comprehensive set, explicitly considering all expert-environment pairings. The size of this polytope adapts naturally to the degree of environment heterogeneity, preventing excessive pessimism when environments are similar while capturing complex variations when environments substantially differ.

where $\boldsymbol{\lambda} = (\lambda_1, \ldots, \lambda_m)$ lies in the probability simplex $\Delta_m = \{ \boldsymbol{\lambda} \in \mathbb{R}^m_{\geq 0} : \sum_{e=1}^m \lambda_e = 1 \}$. The environment-wise risk of a model $f$ on environment $e$ is:

$$
\mathcal{R}_e(\theta) := \mathbb{E}_{(\mathbf{x},y) \sim \mathcal{P}_e} \left[ \ell\big(f_\theta(\mathbf{x}), y\big) \right]. \tag{3}
$$

Hence, GroupDRO solves:

$$
\min_{\theta \in \Theta} \max_{\boldsymbol{\lambda} \in \Delta_m} \sum_{e=1}^m \lambda_e \, \mathcal{R}_e(\theta). \tag{4}
$$

This objective prioritizes the highest-loss environment, driving the model to perform well even under worst-case shifts. In practice, at any training iteration $t$, GroupDRO uses an iterative procedure where *i)* $\mathcal{R}_e(\theta_t)$ is estimated for each environment $e$; *ii)* the weights $\boldsymbol{\lambda}_t$ are updated (e.g., by exponentiated gradient) to emphasize environments with higher risks; *iii)* a gradient step updates $\theta_t$ to reduce the weighted sum of losses. Under mild assumptions (e.g., convexity, bounded gradients, losses, and Lipschitz continuity), convergence occurs at a rate of $\mathcal{O}(1/\sqrt{T})$ [6, 32].

### 3.4 Motivation for extensions

GroupDRO trains a single classifier to be robust against worst-case mixtures of predefined environment distributions. While effective, this single-classifier approach can be restrictive when optimal decision strategies inherently diverge across environments. Furthermore, its uncertainty set, based on evaluating this single strategy across mixtures, may not explicitly isolate or optimize against all distinct classes of vulnerabilities. This motivates exploring extensions that can address these aspects through a more expressive uncertainty framework (see Appendix A.1 for a more detailed discussion).

### 3.5 Proposed method: Multi-Expert Distributionally Robust Optimization (MEDRO)

To this end, we propose MEDRO which extends GroupDRO by enlarging the uncertainty set $\mathcal{Q}$ to handle more potential distribution shifts. Consider a neural net classifier $f : \mathcal{X} \to \mathcal{Y}$ typically decomposed into a shared feature extractor $\phi : \mathcal{X} \to \mathcal{Z}$ and a linear (or shallow) head $\omega : \mathcal{Z} \to \mathcal{Y}$. In MEDRO, we maintain a shared feature extractor $\phi$ but introduce $m$ expert heads, $\omega_1, \ldots, \omega_m$, each tailored to one environment. Let $\theta = \{\phi, \omega_1, \ldots, \omega_m\}$. For a pair $(i, j)$, where expert $i$ is used on data from environment $j$, we define the cross-environment risk as

$$
\mathcal{R}_{i,j}(\theta) = \mathbb{E}_{(\mathbf{x},y) \sim \mathcal{P}_j} \left[ \ell\big((\omega_i \circ \phi)(\mathbf{x}), y\big) \right]. \tag{5}
$$

Rather than tracking just $m$ risks $\{\mathcal{R}_e\}$, MEDRO considers $m^2$ such cross-environment risks $\{\mathcal{R}_{i,j}\}$ serving as candidates for worst-case evaluation. This broader set of risks, corresponding to each potential mismatch between expert $i$ and environment $j$, forms the basis for our expanded uncertainty set (see Figure 1 for a conceptual illustration). This design recognizes that test-time inputs from environment $j$ may be inadvertently labeled or processed as if they came from $i$, especially when domain labels are uncertain.

We assign weights $\lambda_{i,j}$ to each pair $(i,j)$, forming a matrix $\Lambda \in \mathbb{R}^{m \times m}$ that lies in the $(m^2 - 1)$-dimensional probability simplex $\Delta_{m^2} = \left\{ \Lambda \in \mathbb{R}^{m^2}_{\geq 0} \mid \sum_{i=1}^{m} \sum_{j=1}^{m} \lambda_{i,j} = 1 \right\}$. We further introduce a specialization term, $\sum_{i=1}^{m} \mathcal{R}_{i,i}(\theta)$, to ensure expert $i$ remains proficient in environment $i$. Our complete objective is

$$\min_{\theta} \left[ \sum_{i=1}^{m} \mathcal{R}_{i,i}(\theta) \;+\; \gamma \max_{\Lambda \in \Delta_{m^2}} \sum_{i=1}^{m} \sum_{j=1}^{m} \lambda_{i,j}\, \mathcal{R}_{i,j}(\theta) \right], \tag{6}$$

where $\gamma > 0$ balances the importance of cross-environment mismatches. We emphasize that naively enlarging the DRO uncertainty set does not guarantee improved performance and may lead to overly pessimistic models or loss of task-specific performance [6, 33]. MEDRO's design incorporates two key safeguards against these failure modes. First, the worst-case optimization over the expanded uncertainty set remains adaptive and bounded: the $m^2$ cross-environment risks are grounded in the training data and reflect plausible distribution shifts rather than arbitrary worst-case scenarios. When environments exhibit substantial variation, the expanded risk polytope (Figure 1c) captures this heterogeneity, enabling targeted robustness. Conversely, when environments are more homogeneous, the effective uncertainty set contracts naturally, preventing excessive pessimism. Second, the specialization term $\sum_i \mathcal{R}_{i,i}(\theta)$ serves as a critical regularizer that anchors each expert to its native environment, explicitly preserving domain-specific knowledge and preventing the model collapse that could result from overly aggressive robustness optimization. This principled balance between adaptive worst-case protection and specialization anchoring distinguishes MEDRO from naive uncertainty set expansion.

In essence, MEDRO retains the min-max flavor of GroupDRO across $m^2$ risk terms, emphasizing both *native* and *mismatched* expert-environment pairs. Under conditions where $\lambda_{i,j} \approx 0$ for $i \neq j$, the objective of MEDRO simplifies to GroupDRO by focusing on the diagonal terms $\{\mathcal{R}_{i,i}\}$. Conversely, if cross-environment risk is significant ($\lambda_{i,j} > 0$), the expanded uncertainty set can yield enhanced robustness against distribution shifts that a single-environment weighting might fail to capture. Full proofs are provided in Appendix B, showing how this broader formulation encompasses GroupDRO while capturing cross-environment risks.

To solve this objective in (6), we adapt GroupDRO's online approach. At iteration $t$, we (1) estimate $\mathcal{R}_{i,j}(\theta_t)$ with mini-batches from environment $j$, (2) update $\Lambda_{t+1}$ in the $(m^2 - 1)$-dimensional simplex via exponentiated gradient, and (3) update $\theta_{t+1}$ to minimize the weighted sum of cross-environment risks plus the specialization term. Pseudocode is provided in Algorithm 1 of Appendix A.3. This procedure also converges at a rate of $\mathcal{O}(1/\sqrt{T})$, albeit with a factor of $\log(m^2)$ from the higher-dimensional simplex. We provide the full convergence analysis in Appendix C.

**Connection to domain generalization** Although MEDRO is primarily motivated by subpopulation shifts, our theoretical analysis shows that, under the assumption of approximately equal cross-environment risks, the shared feature extractor $\phi^*$ approximately satisfies label-conditional invariance across environments (i.e., $\mathcal{P}_i(Y \mid \phi^*(X)) = \mathcal{P}_j(Y \mid \phi^*(X))$ for all $i,j$). This follows from MEDRO's expanded uncertainty set over all $m^2$ expert–environment pairs, which promotes the removal of environment-specific variations in the learned representation (Appendix D). We empirically evaluate this effect on standard domain-generalization benchmarks in Section 4.4.

## 3.6 Inference with unknown environments

As environment membership is *not* available at test time in our experimental settings, we cannot simply route each sample to its matching expert. Two general strategies are considered for combining the outputs of MEDRO's $m$ expert heads: (1) *simple ensemble*, which averages predictions; and (2) a learned *gating network*, which produces sample-adaptive weights for the experts. Both approaches leverage MEDRO's multiple heads without requiring explicit environment labels at inference.

**Approach 1 (simple ensemble)**  Let each expert $i$ produce a logit vector $(\omega_i \circ \phi)(\mathbf{x}) \in \mathbb{R}^K$ for $i = 1, \ldots, m$. Then, the ensemble logit is:

$$\hat{\mathbf{z}} = \frac{1}{m} \sum_{i=1}^{m} (\omega_i \circ \phi)(\mathbf{x}).$$

Finally, the predicted label is $\hat{y}(\mathbf{x}) = \arg\max_k \hat{z}_k$. This approach is straightforward and requires no additional training.

**Approach 2 (gating network)**  Alternatively, a gating function $g : \mathcal{Z} \to \mathbb{R}^m$ can be employed. This approach is a form of a Mixture of Experts (MoE) model [34, 35, 36], where the gating function learns to assign weights to different experts based on the input. Specifically, the gating function takes the shared features $\phi(\mathbf{x})$ as input and produces logits $(g \circ \phi)(\mathbf{x})$. These logits are then passed through a softmax function to obtain environment-specific gating weights $\alpha(\mathbf{x}) = \mathrm{softmax}\big((g \circ \phi)(\mathbf{x})\big) \in \Delta_m$, where $\Delta_m = \big\{\alpha \in \mathbb{R}_{\geq 0}^m \mid \sum_{i=1}^m \alpha_i = 1\big\}$. Each expert $i$ still produces its logit vector $(\omega_i \circ \phi)(\mathbf{x}) \in \mathbb{R}^K$. The final combined logit vector is then computed as a weighted sum:

$$\hat{\mathbf{z}} = \sum_{i=1}^{m} \alpha_i(\mathbf{x})(\omega_i \circ \phi)(\mathbf{x}),$$

from which the final predicted label $\hat{y}(\mathbf{x}) = \arg\max_k \hat{z}_k$ is derived. The choice between these inference strategies can depend on factors such as the availability of a suitable validation set for training an effective gating mechanism; in its absence, the simple ensemble provides a robust default (see Appendix A.2 for details). In our experiments, we default to the ensemble approach unless specified otherwise.

## 4   Experiments

### 4.1   Experimental design and datasets

Our experimental evaluation uses datasets that span diverse data modalities and distribution shift scenarios to thoroughly evaluate the effectiveness of MEDRO. These datasets are summarized in Table 5 of Appendix E. We selected these datasets for the following key reasons:

1. **Comprehensive coverage of shift types:** Our selection includes both subpopulation shift datasets (Waterbirds [37], CelebA [38], CivilComments [39], MultiNLI [40], MetaShift [41], NICO++ [42], CheXpert [43], Living17 [44]) and domain generalization datasets (Camelyon17 [45], iWildCam [46]), as well as hybrid settings (PovertyMap [47]) that exhibit both types of shifts simultaneously.

2. **Controlled validation:** CelebA and Waterbirds serve as our foundational evaluation environments, following the experimental setup of GroupDRO [6], enabling direct comparison with this established baseline.

3. **Standardized benchmarks:** By leveraging datasets from established benchmarks such as SubpopBench [12] and WILDS [5], we ensure that our evaluation follows rigorous protocols that facilitate fair comparison with state-of-the-art methods.

4. **Varied technical challenges:** These datasets present different technical challenges, from handling high-dimensional image data to processing natural language, and from simple binary classification to multi-class classification with hundreds of categories.

This diversity allows us to evaluate whether MEDRO can provide consistent improvements across different types of distribution shifts, data modalities, and application areas, demonstrating its potential as a general-purpose solution for robust learning.

### 4.2   Method validation on controlled settings

We followed the experimental protocol established in [6], using strong $\ell_2$ regularization to prevent overfitting to majority groups. All methods used identical ResNet-50 architectures and optimizer configurations. Additional training and hyperparameter details are in Appendix E.1.

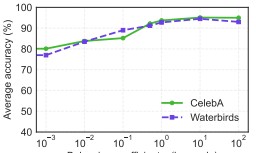 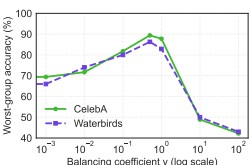 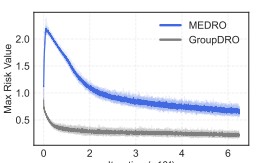 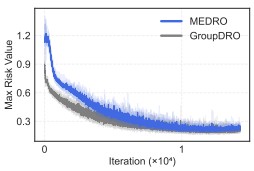

Figure 2: Effect of balancing factor $\gamma$ on CelebA and Waterbirds (log scale). Left: average accuracy; Right: worst-group accuracy.

Figure 3: Worst-case training risk of MEDRO (blue) and GroupDRO (gray), over training iterations. Left: CelebA; Right: Waterbirds.

Table 1: Average and worst-group accuracy (%) on CelebA and Waterbirds. Mean and standard deviation are calculated across five independent runs. The best worst-group accuracy is **boldfaced**.

| CelebA | | | Waterbirds | | |
|---|---|---|---|---|---|
| Method | Avg. | Worst | Method | Avg. | Worst |
| ERM | 95.9 ($\pm$0.1) | 39.2 ($\pm$2.1) | ERM | 95.5 ($\pm$1.5) | 23.5 ($\pm$3.8) |
| GroupDRO | 93.7 ($\pm$0.3) | 85.7 ($\pm$2.2) | GroupDRO | 96.1 ($\pm$1.3) | 84.1 ($\pm$2.4) |
| **MEDRO (ours)** | 92.2 ($\pm$0.8) | **89.4** ($\pm$1.2) | **MEDRO (ours)** | 94.2 ($\pm$1.8) | **86.3** ($\pm$1.9) |

**Balance factor sensitivity analysis** We analyzed how the balancing coefficient $\gamma$ affects model performance in our controlled settings, varying $\gamma$ from $10^{-3}$ to $10^2$. As shown in Figure 2, average accuracy increased with larger $\gamma$ values and saturated near $\gamma = 1$, while worst-group accuracy peaked at $\gamma = 0.5$ and declined for larger values. These results suggest that intermediate $\gamma$ values offer the best trade-off between average and worst-case performance for these experimental conditions.

**Comparing worst-case training risk** Figure 3 illustrates the worst-case training risk, defined as the empirical loss of the most challenging expert-environment pairing at each training iteration for MEDRO, compared to GroupDRO's worst-group risk. MEDRO's explicit consideration of all $m^2$ cross-environment mismatches (i.e., evaluating expert $i$ on environment $j$) means its objective may identify a higher worst-case risk in early training stages, particularly if certain specialized experts initially perform very poorly on non-native environments. This initial emphasis on the most severe of these $m^2$ potential failure modes directly shapes the learning process. It compels the model (both the shared feature extractor $\phi$ and the expert heads $\omega_k$) to prioritize the mitigation of these specific, high-risk vulnerabilities from the outset. As training progresses, MEDRO actively minimizes these challenging mismatch risks. This process cultivates expert heads that are not only proficient in their native domains (due to the specialization term in MEDRO's objective) but also more resilient when applied to mismatched data, ultimately leading to improved robustness on the most challenging subpopulations where such vulnerabilities are critical.

**Results in controlled settings** Under these controlled conditions (Table 1), MEDRO consistently improved worst-group accuracy over GroupDRO (+3.7% on CelebA, +2.2% on Waterbirds) while maintaining comparable average accuracy (e.g., 92.2% vs. 93.7% for GroupDRO on CelebA). This demonstrates MEDRO's ability to substantially close the performance gap on challenging groups with only a minor trade-off in overall average accuracy. These findings suggest MEDRO's explicit consideration of cross-environment risks effectively mitigates spurious correlations and enhances robustness for underrepresented groups.

### 4.3 Large-scale evaluation on SubpopBench

Following the controlled evaluations, we now investigate the robustness of MEDRO under more diverse and challenging subpopulation shifts. To this end, we evaluate on SubpopBench [12], a comprehensive benchmark designed to systematically assess robustness under subpopulation shift across various application areas.

Table 2: Worst-group accuracy (%) on eight SubpopBench datasets, reported as mean $\pm$ standard deviation over three runs. Best results per column are **bolded**.

| Method | Waterbirds | CelebA | CivilC. | MultiNLI | MetaShift | NICO++ | CheXpert | Living17 | Overall |
|---|---|---|---|---|---|---|---|---|---|
| ERM | 69.1 ±4.7 | 62.6 ±1.5 | 63.7 ±1.1 | 66.8 ±0.5 | 82.6 ±0.4 | 37.6 ±2.0 | 50.2 ±3.8 | 28.2 ±1.5 | 57.6 |
| Mixup | 78.2 ±0.4 | 57.8 ±0.8 | 66.1 ±1.3 | 68.5 ±0.6 | 81.0 ±0.8 | 42.7 ±1.4 | 37.4 ±3.5 | 29.8 ±1.8 | 57.7 |
| GroupDRO | 78.6 ±1.0 | 89.0 ±0.7 | 70.6 ±1.2 | **76.0** ±0.7 | 85.6 ±0.4 | 37.8 ±1.8 | 74.5 ±0.2 | 27.2 ±1.5 | 67.4 |
| IRM | 74.5 ±1.5 | 63.0 ±2.5 | 63.2 ±0.8 | 63.6 ±1.3 | 83.0 ±0.1 | 40.0 ±0.0 | 34.4 ±1.7 | 28.2 ±1.5 | 56.2 |
| CVaRDRO | 75.5 ±2.2 | 64.1 ±2.8 | 68.7 ±1.3 | 63.0 ±1.5 | 84.6 ±0.0 | 36.7 ±2.7 | 57.9 ±0.4 | 28.3 ±0.7 | 59.9 |
| JTT | 72.0 ±0.3 | 70.0 ±10.2 | 64.3 ±1.5 | 69.1 ±0.1 | 83.6 ±0.4 | 40.0 ±0.0 | 61.3 ±4.9 | 28.8 ±1.1 | 61.1 |
| LfF | 75.2 ±0.7 | 53.0 ±4.3 | 51.0 ±6.1 | 63.6 ±2.9 | 73.1 ±1.6 | 30.4 ±1.3 | 13.7 ±9.8 | 26.2 ±1.1 | 48.3 |
| LISA | 88.7 ±0.6 | 86.5 ±1.2 | 73.7 ±0.3 | 73.3 ±1.0 | 84.1 ±0.4 | 42.7 ±2.2 | 75.6 ±0.6 | 29.8 ±0.9 | 69.3 |
| MMD | 83.9 ±1.4 | 24.4 ±2.0 | 54.5 ±1.4 | 69.1 ±1.5 | 85.9 ±0.7 | 40.7 ±0.5 | 50.2 ±3.8 | 26.6 ±1.8 | 54.4 |
| ReSample | 77.7 ±1.2 | 87.4 ±0.8 | 73.3 ±0.5 | 72.3 ±0.8 | 85.6 ±0.4 | 40.0 ±0.0 | 75.3 ±0.5 | 30.7 ±2.1 | 67.8 |
| ReWeight | 86.9 ±0.7 | 89.7 ±0.2 | 72.5 ±0.0 | 68.8 ±0.4 | 85.6 ±0.4 | 41.9 ±1.6 | 75.7 ±0.1 | 28.2 ±1.5 | 68.7 |
| SqrtReWeight | 78.6 ±0.1 | 82.4 ±0.5 | 71.7 ±0.4 | 69.5 ±0.7 | 84.6 ±0.7 | 40.0 ±0.0 | 70.0 ±2.3 | 28.2 ±1.5 | 65.6 |
| CBLoss | 86.2 ±0.3 | 89.4 ±0.7 | 73.3 ±0.2 | 72.2 ±0.3 | 85.5 ±0.4 | 37.8 ±1.8 | 74.7 ±0.3 | 28.2 ±1.5 | 68.4 |
| Focal | 71.6 ±0.8 | 59.1 ±2.0 | 62.0 ±1.0 | 69.4 ±0.7 | 81.5 ±0.0 | 36.7 ±2.7 | 42.1 ±4.0 | 28.0 ±1.2 | 56.3 |
| LDAM | 71.0 ±1.8 | 59.6 ±2.4 | 37.4 ±8.1 | 69.6 ±1.6 | 83.6 ±0.4 | 42.0 ±0.9 | 36.4 ±0.3 | 24.7 ±0.8 | 53.0 |
| BSoftmax | 74.1 ±0.9 | 83.3 ±0.5 | 71.2 ±0.4 | 66.9 ±0.4 | 83.1 ±0.7 | 40.4 ±0.3 | 75.4 ±0.5 | 27.5 ±0.8 | 65.2 |
| DFR | **91.0** ±0.3 | 90.4 ±0.1 | 69.6 ±0.2 | 68.5 ±0.2 | 85.4 ±0.4 | 23.7 ±0.7 | 71.7 ±0.2 | 29.0 ±0.2 | 66.2 |
| CRT | 79.7 ±0.3 | 87.2 ±0.3 | 71.1 ±0.1 | 70.7 ±0.1 | 84.1 ±0.4 | **43.3** ±2.7 | 74.6 ±0.3 | **33.9** ±0.1 | 68.1 |
| ReWeightCRT | 78.4 ±0.1 | 87.2 ±0.3 | 71.0 ±0.1 | 69.0 ±0.2 | 85.6 ±0.4 | 23.3 ±1.4 | **76.0** ±0.1 | 33.7 ±0.1 | 65.5 |
| **MEDRO** | 83.8 ±1.3 | 90.4 ±0.4 | 73.4 ±0.5 | 75.7 ±0.8 | 85.9 ±0.4 | 39.1 ±2.5 | 75.0 ±0.4 | 32.6 ±0.7 | 69.5 |
| **MEDRO (w/ gating)** | 84.1 ±1.0 | **91.1** ±0.2 | **74.1** ±0.4 | 75.7 ±0.7 | **87.2** ±0.7 | 39.1 ±2.5 | 75.0 ±0.4 | 33.6 ±0.6 | **70.0** |

**Experimental setup**  We evaluated MEDRO on eight representative SubpopBench tasks spanning diverse modalities. Following the official benchmark protocol, we used worst-group accuracy as the primary metric. Full experimental details and baseline descriptions are provided in Appendix E.2.

**Results on SubpopBench**  Table 2 reports the worst-group accuracy for all methods across the eight SubpopBench datasets. Consistent with previous findings [12], many existing robustness methods exhibit strong performance on specific datasets but do not generalize their effectiveness universally across the varied subpopulation shift scenarios present in the benchmark. In this challenging evaluation, MEDRO (ours) achieved the highest overall worst-group accuracy of 69.5% when averaged across all eight tasks. This leading performance underscores MEDRO's ability to effectively address a diverse range of subpopulation shifts. We attribute this strong generalization to its expanded risk structure, which explicitly considers $m^2$ cross-environment risks, allowing it to better identify and mitigate vulnerabilities beyond conventional worst-case weighting over groups. Furthermore, when augmented with a lightweight gating mechanism for inference (MEDRO w/ gating, as described in Section 3.6), the overall worst-group accuracy improved further to 70.0%, demonstrating the potential for adaptive expert combination at test time.

**Isolating the effect of multi-head architecture**  Since MEDRO employs multiple expert heads, a natural concern is whether its improvements arise from the expanded uncertainty formulation or from ensemble effects. To isolate these factors, we evaluated GroupDRO with a multi-head architecture—a controlled baseline that maintains the same architectural structure but trains each head independently on the standard GroupDRO objective, differing only in random initialization. Both methods use identical ensemble inference at test time. As detailed in Appendix E.2.6, across eight SubpopBench datasets, GroupDRO with multi-head architecture achieves 67.8% overall worst-group accuracy, a +0.4% improvement over single-head GroupDRO (67.4%). MEDRO achieves 69.5%, a +1.7% improvement over the multi-head variant. This pattern indicates that MEDRO's gains stem primarily from modeling cross-environment risks through its expanded $m^2$ uncertainty set rather than from architectural choices.

## 4.4 Evaluation under domain shifts

While MEDRO was initially designed primarily with subpopulation shifts in mind, we now investigate its effectiveness in broader domain generalization scenarios, where shifts between training and test distributions can be more substantial. To this end, we evaluated MEDRO on three representative tasks from the WILDS benchmark [5]: Camelyon17 (histopathology image classification), iWildCam (wildlife camera trap image classification), and PovertyMap (satellite image regression for poverty prediction). These datasets are characterized by more fundamental and structural differences between

Table 3: Performance on WILDS benchmarks, reported as mean $\pm$ standard deviation. (a) Camelyon17: OOD validation and test accuracy (%) from 10 independent runs. (b) iWildCam: OOD validation and test macro F1 scores (%) from 3 independent runs. The best result in each column is **bolded**. A dash indicates missing results that were unavailable in the original benchmark study [18].

| Method | (a) Camelyon17 | | (b) iWildCam | |
|---|---|---|---|---|
| | Validation | Test | Validation | Test |
| ERM (scratch) | 84.9 ($\pm$3.1) | 70.8 ($\pm$7.2) | - | - |
| ERM (ImageNet) | 91.3 ($\pm$0.2) | 84.2 ($\pm$2.1) | **37.4** ($\pm$1.3) | 31.0 ($\pm$1.3) |
| CORAL | 86.2 ($\pm$1.4) | 59.5 ($\pm$7.7) | 37.0 ($\pm$1.2) | **32.8** ($\pm$0.1) |
| IRM | 86.2 ($\pm$1.4) | 64.2 ($\pm$8.1) | 20.2 ($\pm$7.6) | 15.1 ($\pm$4.9) |
| GroupDRO | 85.5 ($\pm$2.4) | 68.4 ($\pm$7.3) | 26.3 ($\pm$0.2) | 23.9 ($\pm$2.1) |
| DANN | - | - | - | 31.9 ($\pm$1.4) |
| VREx | 82.3 ($\pm$1.3) | 71.5 ($\pm$8.3) | - | - |
| LISA | 81.8 ($\pm$1.3) | 77.1 ($\pm$6.5) | - | - |
| Fish | 82.5 ($\pm$1.2) | 79.5 ($\pm$6.0) | 25.8 ($\pm$0.5) | 24.2 ($\pm$0.9) |
| SWAD | 88.1 ($\pm$1.5) | 83.9 ($\pm$0.9) | 31.6 ($\pm$0.2) | 29.1 ($\pm$0.1) |
| L2A-OT | 86.3 ($\pm$3.4) | 77.5 ($\pm$6.7) | 22.8 ($\pm$2.9) | 18.1 ($\pm$3.2) |
| HeckmanDG | 90.6 ($\pm$2.4) | 87.3 ($\pm$2.4) | 34.5 ($\pm$0.9) | 31.8 ($\pm$0.3) |
| **MEDRO (ours)** | **92.6** ($\pm$0.5) | **87.8** ($\pm$1.9) | 34.1 ($\pm$0.7) | 31.5 ($\pm$0.8) |

training and test environments compared to the group-level variations typically seen in subpopulation shift problems. Theoretical motivations for domain generalization are provided in Appendix D.

**Experimental setup** Following the official WILDS protocol, we used the designated out-of-distribution (OOD) validation sets for model selection and reported performance on the OOD test sets using dataset-specific metrics (accuracy for Camelyon17, macro F1 for iWildCam, and Pearson correlation for PovertyMap). All baseline results were taken from prior work that adheres to the WILDS codebase and evaluation guidelines [18]. Additional experimental details are provided in Appendix E.3.

**Results on WILDS** The performance of MEDRO and baseline methods on these challenging domain generalization tasks is presented in Table 3 and Table 4. On Camelyon17 (Table 3a), MEDRO achieves the best OOD validation (92.6%) and test (87.8%) performance, surpassing all listed baselines, including GroupDRO and other strong domain generalization methods. For iWildCam (Table 3b), MEDRO (31.5% test F1) again substantially outperforms GroupDRO and remains highly competitive. In the regression task on PovertyMap (Table 4), using a linear regressor head, MEDRO demonstrates strong results. Its average Pearson correlation (0.80 test) is competitive with top-performing methods and notably surpasses GroupDRO. For worst-group correlation, MEDRO (0.49 test) shows significant improvement over GroupDRO and is second only to HeckmanDG. Across these diverse WILDS benchmarks, MEDRO consistently delivers substantial improvements over GroupDRO. These results indicate its expanded uncertainty set, which considers specific expert-environment mismatches, is beneficial for tackling complex domain shifts and enhances out-of-distribution performance, complementing existing domain generalization strategies.

## 5 Conclusion

In this work, we investigated the limitations of single-classifier GroupDRO in handling heterogeneous environments and proposed MEDRO, which significantly broadens the uncertainty set to include cross-environment mismatches. By assigning individual heads to each environment but sharing a common feature extractor, MEDRO retains the core adversarial weighting principle of GroupDRO while offering greater flexibility to specialize across a range of subpopulations or domains. Our empirical results on both subpopulation-shift and domain-shift benchmarks confirm that this approach substantially improves test-time robustness, especially in settings with strong spurious correlations or large domain discrepancies.

Table 4: Performance comparison on the PovertyMap dataset from WILDS benchmark, reported as the Pearson correlation coefficient (mean $\pm$ std over five runs) on both the OOD validation and test sets. Experiments use the original 5-fold dataset splits provided in WILDS. The best result in each column is **bolded**. We do not report worst-group performance for 'Fish', because it has not been reported in [23].

| Method | Average Corr. | | Worst-Group Corr. | |
|---|---|---|---|---|
| | Validation | Test | Validation | Test |
| ERM | 0.80 ($\pm$0.04) | 0.78 ($\pm$0.03) | 0.51 ($\pm$0.06) | 0.45 ($\pm$0.06) |
| CORAL | 0.80 ($\pm$0.04) | 0.77 ($\pm$0.05) | 0.52 ($\pm$0.06) | 0.44 ($\pm$0.06) |
| IRM | **0.81** ($\pm$0.03) | 0.77 ($\pm$0.05) | **0.53** ($\pm$0.06) | 0.43 ($\pm$0.07) |
| GroupDRO | 0.78 ($\pm$0.05) | 0.75 ($\pm$0.07) | 0.46 ($\pm$0.07) | 0.39 ($\pm$0.06) |
| DANN | 0.77 ($\pm$0.04) | 0.69 ($\pm$0.04) | 0.44 ($\pm$0.11) | 0.33 ($\pm$0.10) |
| Fish | **0.81** ($\pm$0.01) | **0.81** ($\pm$0.01) | – | – |
| SWAD | 0.78 ($\pm$0.03) | 0.77 ($\pm$0.04) | 0.48 ($\pm$0.09) | 0.45 ($\pm$0.11) |
| HeckmanDG | **0.81** ($\pm$0.03) | **0.81** ($\pm$0.03) | **0.53** ($\pm$0.06) | **0.51** ($\pm$0.04) |
| **MEDRO (ours)** | 0.80 ($\pm$0.04) | 0.80 ($\pm$0.03) | 0.50 ($\pm$0.06) | 0.49 ($\pm$0.03) |

In addition, we showed that MEDRO naturally accommodates situations where environment labels are unavailable at inference by employing either a simple ensemble of experts or a learned gating mechanism. These test-time strategies enable practical deployment without the need to know the environment membership of new samples. Moving forward, we see multiple directions for advancing this methodology: examining semi-supervised or dynamically evolving environments, and exploring theoretical questions regarding the minimal conditions under which expert heads significantly enhance robustness. Overall, our findings affirm that environment-specific parameterization is a compelling way to address complex, real-world distribution shifts, offering a principled balance between worst-case robustness and overall accuracy.

## Acknowledgements

This research was supported by Brain Korea 21 FOUR, the Ministry of Science and ICT (MSIT) in Korea under the ITRC support program supervised by the Institute for Information Communication Technology Planning and Evaluation (IITP-2020-0-01749), a Korea University grant, and the National Research Foundation of Korea grant funded by the Korea government (RS-2022-00144190).

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

# A Supplementary Details for Multi-Expert DRO (MEDRO)

## A.1 Rationale for the expanded uncertainty set of MEDRO

Distributional robustness hinges on what set of risks the learner treats as plausible test–time failures. GroupDRO considers the $m$ risks $\{R_e\}_{e=1}^m$ of a single classifier across the $m$ training environments and safeguards against their worst-case mixture. This is effective when a common decision rule can serve all environments, but becomes brittle whenever the Bayes-optimal strategies diverge.

MEDRO remedies this by (i) allocating an expert $\omega_i \circ \phi$ to each environment $\mathcal{P}_i$ and (ii) enlarging the uncertainty set to the $m^2$ cross-environment risks:

$$\mathcal{R}_{i,j}(\theta) := \mathbb{E}_{(\mathbf{x},y)\sim\mathcal{P}_j}[\ell((\omega_i \circ \phi)(\mathbf{x}), y)], \quad 1 \le i, j \le m.$$

These risks quantify specialization (the diagonal $\mathcal{R}_{i,i}$) and mismatch (the off-diagonal $\mathcal{R}_{i,j}$) in a single view. The min-max term in Eq. (6) therefore searches the convex hull of all $m^2$ points—visually the shaded polytope in Figure 1c—instead of the simple in Figure 1a. Because the former strictly contains the latter, MEDRO does not underestimate worst-case error and reduces to GroupDRO when $\lambda_{i,j} = 0$ for $i \ne j$.

A toy illustration helps ground this expansion. In the Waterbirds dataset, an expert trained on *land* backgrounds (i.e., environment $e_{\text{land}}$) may rely on background cues that are spurious for *water* birds. If at test time, a water-background image were mis-routed to the land expert, the loss could spike despite GroupDRO's guarantee. MEDRO penalizes exactly this scenario through $\mathcal{R}_{\text{land,water}}$, compelling the shared encoder $\phi$ either to discard the cue or to make the experts mutually resilient.

Finally, the additional specialization term $\sum_i \mathcal{R}_{i,i}(\theta)$ keeps each expert near its environment's Bayes risk, preventing the trivial solution where all heads collapse into an identical, over-regularized classifier. The balance hyperparameter $\gamma$ mediates this trade-off and is studied empirically in Section 4.2 (Figure 2).

In summary, MEDRO's uncertainty set is deliberately broader, capturing both per-environment optimality and cross-expert vulnerabilities, while retaining the computational structure of GroupDRO.

## A.2 Gating network for inference: training details

The gating network $g$ can, in principle, be trained end-to-end with the main MEDRO objective, a common strategy in MoE literature [35, 36], or trained separately on a held-out validation set with known environment labels. In our experiments, we adopted the latter approach, training the gating network on a dedicated validation set. This staged training approach, while potentially suboptimal compared to joint optimization, is often employed for practical reasons such as simplifying the training process or when expert models are pre-trained [48, 49]. For the gate to learn effectively, this validation set should ideally be representative of the various environments. For instance, in subpopulation shift problems, having roughly balanced representation of subgroups within this validation data may be beneficial, although this constitutes a practical consideration regarding data availability. While a sufficiently large and balanced validation set is ideal, a relatively small number of labeled examples per environment might still suffice to train a useful gate. We opted not to explore end-to-end training of the gating network concurrently with the MEDRO objective due to the anticipated complexities in the optimization dynamics arising from such joint training.

## A.3 MEDRO Training Algorithm

---

**Algorithm 1** Multi-Expert Distributionally Robust Optimization (MEDRO) Training Procedure

---

 1: **Require:** Training data from $m$ environments $\{\mathcal{D}_e \sim \mathcal{P}_e\}_{e=1}^m$
 2: **Require:** Balance factor $\gamma > 0$
 3: **Require:** Learning rate for model parameters $\eta_\theta$
 4: **Require:** Learning rate for risk weights $\eta_\Lambda$
 5: **Require:** Number of training iterations (batch steps) $T$
 6: **Initialize:** Model parameters $\theta_0 = \{\phi_0, \omega_{1,0}, \ldots, \omega_{m,0}\}$
 7: **Initialize:** Risk weights $\Lambda_0 = \{(\lambda_0)_{i,j}\}$ (e.g., uniform: $(\lambda_0)_{i,j} = 1/m^2$ for all $i, j$)
 8: **for** $t = 0, \ldots, T-1$ **do**
 9:    *# Step 1: Estimate all cross-environment risks*
10:    **for** $i = 1, \ldots, m$ **do**
11:      **for** $j = 1, \ldots, m$ **do**
12:        Sample a mini-batch $B_j \sim \mathcal{P}_j$ from environment $j$.
13:        Estimate $\hat{\mathcal{R}}_{i,j}(\theta_t) = \frac{1}{|B_j|} \sum_{(\mathbf{x},y) \in B_j} \ell\big((\omega_{i,t} \circ \phi_t)(\mathbf{x}), y\big)$.
14:      **end for**
15:    **end for**
16:    *# Step 2: Update risk weights $\Lambda$*
17:    Let $R_t = \{\hat{\mathcal{R}}_{i,j}(\theta_t)\}_{i,j=1}^m$.
18:    Update $\Lambda_{t+1}$ from $\Lambda_t$ to emphasize higher risks in $R_t$.  *# e.g., using exponentiated gradient*
19:    **for** $i = 1, \ldots, m$ **do**
20:      **for** $j = 1, \ldots, m$ **do**
21:        $(\lambda'_{t+1})_{i,j} \leftarrow (\lambda_t)_{i,j} \exp(\eta_\Lambda \cdot \hat{\mathcal{R}}_{i,j}(\theta_t))$.
22:      **end for**
23:    **end for**
24:    Normalize $\Lambda_{t+1}$: $(\lambda_{t+1})_{i,j} \leftarrow (\lambda'_{t+1})_{i,j} / \sum_{a,b} (\lambda'_{t+1})_{a,b}$.
25:    *# Step 3: Update model parameters $\theta$*
26:    Define the loss for parameter update using $\Lambda_{t+1}$:
27:    $L_{\text{MEDRO}}(\theta_t, \Lambda_{t+1}) = \sum_{k=1}^m \hat{\mathcal{R}}_{k,k}(\theta_t) + \gamma \sum_{i=1}^m \sum_{j=1}^m (\lambda_{t+1})_{i,j} \hat{\mathcal{R}}_{i,j}(\theta_t)$.
28:    Compute gradient $g_t = \nabla_\theta L_{\text{MEDRO}}(\theta_t, \Lambda_{t+1})$.
29:    Update parameters: $\theta_{t+1} \leftarrow \theta_t - \eta_\theta g_t$. *# This involves updating $\phi_{t+1}$ and all $\omega_{k,t+1}$*
30: **end for**
31: **Return:** Trained parameters $\theta_T$.

---

# B    Containment analysis of GroupDRO in MEDRO

In this appendix, we provide an argument that our proposed MEDRO formulation either exactly or approximately contains the standard GroupDRO objective. We show how GroupDRO's worst-case weighting over $m$ environments can be embedded in our $(m^2 - 1)$-dimensional simplex, ensuring that MEDRO subsumes GroupDRO as a special (or near-special) case, even when the diagonal risk $\mathcal{R}_{i,i}$ in MEDRO does not perfectly match the single-classifier risk $\mathcal{R}_i$ from GroupDRO.

## B.1    Revisiting GroupDRO and MEDRO

**GroupDRO**    Let there be $m$ environments $\{e_1, \ldots, e_m\}$, each associated with a distribution $P_e$. The GroupDRO objective is

$$\min_{\theta_G \in \Theta_G} \max_{\boldsymbol{\lambda} \in \Delta_m} \sum_{e=1}^m \lambda_e \mathcal{R}_e(\theta_G), \tag{7}$$

where $\Delta_m$ is the $(m-1)$-dimensional simplex, and $\mathcal{R}_e(\theta_G)$ is the expected loss under environment $e$ with parameters $\theta_G$.

**MEDRO**    In our approach, we no longer rely on a single set of parameters $\theta_G$. Instead, we introduce a shared parameter $\phi$, and environment-specific heads $\{\omega_1, \ldots, \omega_m\}$, one per environment. We

combine them as $\theta = \{\phi, \omega_1, \ldots, \omega_m\}$. This entire set of parameters defines a multi-head model. The MEDRO objective is

$$\min_{\theta} \left[ \sum_{i=1}^{m} \mathcal{R}_{i,i}(\theta) \; + \; \gamma \max_{\Lambda \in \Delta_{m^2}} \sum_{i=1}^{m} \sum_{j=1}^{m} \lambda_{i,j} \, \mathcal{R}_{i,j}(\theta) \right], \tag{8}$$

where $\Delta_{m^2} = \left\{ \Lambda \in \mathbb{R}_{\geq 0}^{m^2} \mid \sum_{i=1}^{m} \sum_{j=1}^{m} \lambda_{i,j} = 1 \right\}$ is the probability simplex of dimension $(m^2 - 1)$.

**Parameter space relationship**   An important connection between GroupDRO and MEDRO exists when we constrain all classifier heads in MEDRO to be identical ($\omega_1 = \omega_2 = \cdots = \omega_m = \omega$). In this restricted case, our MEDRO model with parameters $\theta = \{\phi, \omega, \ldots, \omega\}$ becomes functionally equivalent to a GroupDRO model with parameters $\theta_G = \{\phi, \omega\}$. This equivalence allows us to compare $\mathcal{R}_{i,i}(\theta)$ with $\mathcal{R}_i(\theta_G)$ in following analysis.

## B.2   Embedding GroupDRO's weights into $(m^2 - 1)$ dimensions

A key step in showing that MEDRO can replicate GroupDRO's worst-case solution is to embed any weighting $\boldsymbol{\lambda} = \{\lambda_1, \ldots, \lambda_m\} \in \Delta_m$ into $\Lambda \in \Delta_{m^2}$ by placing all mass on the diagonal entries:

$$\lambda_{i,j} = \begin{cases} \lambda_i & \text{if } i = j, \\ 0 & \text{otherwise.} \end{cases} \tag{9}$$

Since $\sum_{i,j} \lambda_{i,j} = \sum_{j}^{m} \lambda_j = 1$, indeed $\Lambda$ lies in $\Delta_{m^2}$. Moreover, for any function $F_{i,j}$,

$$\sum_{i=1}^{m} \sum_{j=1}^{m} \lambda_{i,j} F_{i,j} = \sum_{j=1}^{m} \lambda_j F_{j,j}.$$

This diagonal embedding is crucial as it enables MEDRO's $\max_{\Lambda}$ optimization to directly capture GroupDRO's $\max_{\boldsymbol{\lambda}}$ objective when focusing on diagonal elements.

## B.3   Exact containment if $\mathcal{R}_{i,i}(\theta) = \mathcal{R}_i(\theta_G)$

If each diagonal risk in MEDRO equals the single-classifier risk from GroupDRO, we obtain exact containment.

**Exact case**   Consider the setting where MEDRO parameter $\theta = \{\phi, \omega, \ldots, \omega\}$ have identical classifier heads, corresponding to GroupDRO parameters $\theta_G = \{\phi, \omega\}$. Suppose that $\mathcal{R}_{i,i}(\theta) = \mathcal{R}_i(\theta_G)$ for each $i$ in this setting. Then,

$$\max_{\Lambda \in \Delta_{m^2}} \sum_{i=1}^{m} \sum_{j=1}^{m} \lambda_{i,j} \mathcal{R}_{i,j}(\theta) \; \geq \; \max_{\boldsymbol{\lambda} \in \boldsymbol{\Delta}_m} \sum_{j=1}^{m} \lambda_j \mathcal{R}_{j,j}(\theta) \; = \; \max_{\boldsymbol{\lambda} \in \Delta_m} \sum_{j=1}^{m} \lambda_j \mathcal{R}_j(\theta_G), \tag{10}$$

where the last equality follows from the assumption that $\mathcal{R}_{j,j}(\theta) = \mathcal{R}_j(\theta_G)$. Minimizing over $\theta$ shows that MEDRO's min-max objective is at most the GroupDRO optimum, hence MEDRO includes GroupDRO as a subproblem.

## B.4   Approximate containment if $\left| \mathcal{R}_{i,i}(\theta) - \mathcal{R}_i(\theta_G) \right| \leq \varepsilon_i$

While the exact containment provides a clean theoretical guarantee when $\mathcal{R}_{i,i}(\theta) = \mathcal{R}_i(\theta_G)$, in practice there may be a small discrepancy even when using identical classifier heads. We now analyze how MEDRO approximates GroupDRO when these risks are not exactly equal.

**Approximate case**   Suppose that $\left| \mathcal{R}_{i,i}(\theta) - \mathcal{R}_i(\theta_G) \right| \leq \varepsilon_i$ for $i = 1, \ldots, m$ when using equivalent parameter configurations. Let $\boldsymbol{\lambda}^*$ be the optimal worst-case weighting for GroupDRO. We embed $\boldsymbol{\lambda}^*$

into $\Lambda^*$ via diagonal entries as in (9). Then

$$
\begin{aligned}
\sum_{i,j} \lambda_{i,j}^* \mathcal{R}_{i,j}(\theta) &= \sum_{j=1}^m \lambda_j^* \mathcal{R}_{j,j}(\theta) \\
&\geq \sum_{j=1}^m \lambda_j^* \big[ \mathcal{R}_j(\theta_G) - \varepsilon_j \big] \\
&= \sum_{j=1}^m \lambda_j^* \mathcal{R}_j(\theta_G) - \sum_{j=1}^m \lambda_j^* \varepsilon_j \\
&\geq \sum_{j=1}^m \lambda_j^* \mathcal{R}_j(\theta_G) - \sum_{j=1}^m \lambda_j^* \max_k \varepsilon_k \\
&= \sum_{j=1}^m \lambda_j^* \mathcal{R}_j(\theta_G) - \max_j \varepsilon_j,
\end{aligned}
$$

because $\sum_{j=1}^m \lambda_j^* = 1$. Hence MEDRO approximates GroupDRO to within $O(\max_j \varepsilon_j)$, providing a precise bound on the approximation error.

## B.5 Empirical illustration: GroupDRO-MEDRO continuum

The theoretical analysis above shows that MEDRO contains GroupDRO as a special case when expert heads are constrained to be identical. To illustrate this relationship empirically, we conducted a preliminary experiment on Waterbirds.

We added a regularization term to MEDRO that penalizes expert dissimilarity: $\mathcal{C} \cdot \frac{1}{m} \sum_{i=1}^m \|\omega_i - \bar{\omega}\|_2$, where $\bar{\omega}$ is the average parameter vector across all $m$ expert heads. As $\mathcal{C}$ increases, this penalty drives experts toward similar parameters. Small $\mathcal{C}$ values allow expert specialization (MEDRO-like), while large $\mathcal{C}$ values enforce similarity (GroupDRO-like).

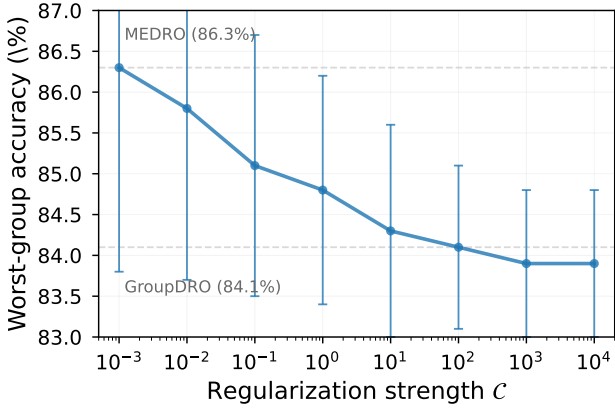

Figure 4: Worst-group test accuracy on Waterbirds as a function of constraint strength $\mathcal{C}$. Error bars represent standard deviation across five runs.

As shown in Figure 4, performance exhibits a power-law decay from 86.3% ($\mathcal{C} = 0.001$) to 83.9% ($\mathcal{C} \geq 1000$). At high constraint levels, performance converges to the GroupDRO baseline (84.1% from Table 1), consistent with the theoretical prediction. The 2.4 percentage point gap between the two extremes suggests that expert specialization provides measurable benefits on this dataset, though evaluation across multiple datasets would be needed to characterize this effect more broadly.

# C  Convergence analysis of MEDRO

## C.1  Problem formulation and assumptions

As established in Section 3, MEDRO operates in an expanded uncertainty set compared to GroupDRO. Considering $m^2$ cross-environment risks rather than $m$ environment risks. Recall that MEDRO involves optimizing over a shared parameter $\phi$ and environment-specific experts $\{\omega_1, \ldots, \omega_m\}$, with the objective including both a specialization term and a worst-case cross-environment robustness term.

**Saddle-point reformulation**  Let $\theta := \{\phi, \omega_1, \ldots, \omega_m\}$ collectively denote all parameters of the model. By introducing randomness $\xi$ (e.g., stochastic mini-batches) in the data sampling, we can write the expected objective in min-max (saddle-point) form:

$$\min_\theta \max_{\Lambda \in \Delta_{m^2}} \mathbb{E}_\xi[L(\theta, \Lambda, \xi)], \tag{11}$$

where $L(\theta, \Lambda, \xi) = \sum_{i=1}^m \mathcal{R}_{i,i}(\theta, \xi) + \gamma \sum_{i=1}^m \sum_{j=1}^m \lambda_{i,j} \mathcal{R}_{i,j}(\theta, \xi)$.

**Assumptions**

1. **Convexity / Concavity:** $\mathcal{R}_{i,j}(\theta, \xi)$ is convex in $\theta$ and $\Lambda \mapsto -\sum_{i,j} \lambda_{i,j} \mathcal{R}_{i,j}(\theta, \xi)$ is concave over $\Delta_{m^2}$.
2. **Compactness:** The parameter set $\theta$ is compact, $\|\theta\| \leq B_\theta$, and $\Delta_{m^2}$ is a simplex, whose geometry yields a $\log(m^2)$ term in mirror descent.
3. **Lipschitz gradients:** $\|\nabla_\theta \mathcal{R}_{i,j}(\theta, \xi)\| \leq B_\nabla$ implies $\|g_t\| \leq B_\nabla$, and $\|l_t\|_\infty \leq \gamma B_l$ on the dual side, where $l_t = \{\gamma \mathcal{R}_{i,j}(\theta_t, \xi_t)\}_{1 \leq i,j \leq m}$. See Eq. (12) for a formal definition of $g_t$.

Under these assumptions, we show an $\mathcal{O}(1/\sqrt{T})$ convergence rate under an online mirror descent procedure.

## C.2  Online mirror descent algorithm

**Gradient definitions**  At iteration $t$, we observe a stochastic mini-batch $\xi_t$. We define:

$$g_t = \nabla_\theta \left[ \sum_{i=1}^m \mathcal{R}_{i,i}(\theta_t, \xi_t) + \gamma \sum_{i=1}^m \sum_{j=1}^m [\Lambda_t]_{i,j} \mathcal{R}_{i,j}(\theta_t, \xi_t) \right], \tag{12}$$

where $[\Lambda_t]_{i,j}$ denotes the $(i,j)$-th element of the weight matrix $\Lambda_t$ at iteration $t$. The primal gradient $g_t$ is w.r.t. $\theta$, and the dual update uses $l_t$ (a matrix of scaled risks).

**Primal update (mirror descent)**  We update $\theta$ by a mirror descent step:

$$\theta_{t+1} = \arg\min_\theta \{\eta_\theta \langle g_t, \theta \rangle + D_\psi(\theta, \theta_t)\}, \tag{13}$$

where $\eta_\theta \geq 0$ is the primal learning rate, and $D_\psi$ is a Bregman divergence measuring the distance between $\theta$ and $\theta_t$.

**Dual update (mirror ascent)**  Simultaneously, we update the dual variable $\Lambda \in \Delta_{m^2}$ by a mirror-ascent step:

$$\Lambda_{t+1} = \arg\max_{\Lambda \in \Delta_{m^2}} \{\eta_\Lambda \langle l_t, \Lambda \rangle - D_\nu(\Lambda, \Lambda_t)\}, \tag{14}$$

where $\eta_\Lambda \geq 0$ is the dual learning rate and $D_\nu$ is a Bregman divergence on the $\Delta_{m^2}$ simplex. This extends GroupDRO's dual update to our expanded uncertainty set.

The dual updates operate over an $(m^2 - 1)$-dimensional simplex rather than the $(m-1)$-dimensional simplex in GroupDRO. Repeating this procedure for $t = 1, 2, \ldots, T$ yields sequences $\{\theta_t, \Lambda_t\}$.

## C.3  Regret analysis

Our convergence analysis follows the approach in [6], which builds upon the stochastic approximation framework of [50]. Let $\theta^*$ and $\Lambda^*$ be a saddle point of the min-max objective.

**Regret definitions**   Define the cumulative regrets for primal and dual variables:

$$R_T^\theta = \sum_{t=1}^T \langle g_t, \theta_t - \theta^* \rangle, \quad R_T^\Lambda = \sum_{t=1}^T \langle l_t, \Lambda^* - \Lambda_t \rangle, \tag{15}$$

where $\langle l_t, \Lambda \rangle = \sum_{i=1}^m \sum_{j=1}^m [l_t]_{i,j} [\Lambda]_{i,j}$ represents the element-wise product sum between matrices.

**Primal regret bound**   From mirror descent theory for saddle-point optimization [50], if $\|\theta\| \le B_\theta$ and $\|g_t\| \le B_\nabla$, then

$$\mathbb{E}[R_T^\theta] = \mathbb{E}\left[\sum_{t=1}^T \langle g_t, \theta_t - \theta^* \rangle\right] \tag{16}$$

$$\le \frac{B_\theta^2}{\eta_\theta} + \frac{\eta_\theta}{2} \sum_{t=1}^T \mathbb{E}[\|g_t\|^2]. \tag{17}$$

For simplicity, assume $\mathbb{E}[\|g_t\|^2] \le B_\nabla^2$. Then

$$\mathbb{E}[R_T^\theta] \le \frac{B_\theta^2}{\eta_\theta} + \frac{\eta_\theta T B_\nabla^2}{2}. \tag{18}$$

**Dual regret bound**   For mirror ascent on the simplex $\Delta_{m^2}$, If $\|l_t\|_\infty \le \gamma B_l$, the standard regret bound gives

$$\mathbb{E}[R_T^\Lambda] = \mathbb{E}\left[\sum_{t=1}^T \langle l_t, \Lambda^* - \Lambda_t \rangle\right] \tag{19}$$

$$\le \frac{\log(m^2)}{\eta_\Lambda} + \frac{\eta_\Lambda T [\gamma B_l]^2}{2}. \tag{20}$$

The $\log(m^2)$ term arises from the maximum divergence between any two points in the $(m^2 - 1)$-dimensional simplex.

**Total regret bound**   Summing the two regrets:

$$\mathbb{E}[R_T] = \mathbb{E}[R_T^\theta + R_T^\Lambda] \tag{21}$$

$$\le \frac{B_\theta^2}{\eta_\theta} + \frac{\log(m^2)}{\eta_\Lambda} + \frac{\eta_\theta T B_\nabla^2}{2} + \frac{\eta_\Lambda T \gamma^2 B_l^2}{2}. \tag{22}$$

**Optimal learning rate**   The optimal learning rates that minimize this bound are:

$$\eta_\theta^* = \sqrt{\frac{2 B_\theta^2}{T B_\nabla^2}}, \quad \eta_\Lambda^* = \sqrt{\frac{2 \log(m^2)}{T \gamma^2 B_l^2}}. \tag{23}$$

Substituting these optimal learning rates:

$$\mathbb{E}[R_T] \le \sqrt{2 T B_\theta^2 B_\nabla^2} + \sqrt{2 T \log(m^2) \gamma^2 B_l^2} = \mathcal{O}(\sqrt{T}). \tag{24}$$

Dividing by $T$ shows the average regret is $\mathcal{O}(1/\sqrt{T})$.

## C.4   Convergence rate

Given the bound on the average regret:

$$\frac{\mathbb{E}[R_T]}{T} = \frac{1}{T} \mathbb{E}[R_T^\theta + R_T^\Lambda] = \mathcal{O}\left(\frac{1}{\sqrt{T}}\right), \tag{25}$$

we can define the average iterates $\bar{\theta}_T = \frac{1}{T}\sum_{t=1}^T \theta_t$ and $\bar{\Lambda}_T = \frac{1}{T}\sum_{t=1}^T \Lambda_t$. By Jensen's inequality applied to the convex-concave objective, the expected suboptimality at these average iterates is bounded by the average regret:

$$\mathbb{E}[L(\bar{\theta}_T, \Lambda^*) - L(\theta^*, \bar{\Lambda}_T)] \leq \frac{\mathbb{E}[R_T]}{T} = \mathcal{O}\left(\frac{1}{\sqrt{T}}\right). \tag{26}$$

This establishes that MEDRO achieves the same convergence rate of $\mathcal{O}(1/\sqrt{T})$ as standard Group-DRO, despite expanded parameter and uncertainty spaces. The $log(m^2)$ term in the dual regret bound introduces a constant factor of 2 compared to the $log(m)$ term in GroupDRO.

# D   Theoretical connection between MEDRO objective, expert optimality, and feature invariance

This subsection lays the theoretical groundwork for understanding how the Multi-Expert Distributionally Robust Optimization (MEDRO) objective facilitates domain generalization. We begin by defining key terms and then explore the relationship between the objective function, the optimality of individual environment experts, and the emergence of domain-invariant feature representations under specific conditions.

## D.1   Preliminaries and definitions

We consider a domain generalization setting with $m$ distinct source domains, each with a data distribution $\mathcal{P}_e$ over the input-output space $\mathcal{X} \times \mathcal{Y}$, for $e \in \{1, 2, \ldots, m\}$. Our model, parameterized by $\theta = \{\phi, \omega_1, \ldots, \omega_m\}$, consists of a shared feature extractor $\phi : \mathcal{X} \to \mathcal{Z}$ and $m$ environment-specific "expert" heads $\omega_k : \mathcal{Z} \to \mathcal{Y}$ for $k \in \{1, \ldots, m\}$.

The cross-environment risk for expert head $i$ evaluated on data from environment $j$ is defined as:

$$\mathcal{R}_{i,j}(\theta) = \mathbb{E}_{(\mathbf{x},y)\sim\mathcal{P}_j}\left[\ell((\omega_i \circ \phi)(\mathbf{x}), y)\right] \tag{27}$$

where $\ell(\cdot, \cdot)$ is a given loss function. The MEDRO objective function to be minimized is:

$$L(\theta) = \sum_{k=1}^m \mathcal{R}_{k,k}(\theta) + \gamma \max_{i,j\in\{1,\ldots,m\}} \mathcal{R}_{i,j}(\theta) \tag{28}$$

where $\gamma > 0$ is a hyperparameter balancing native-environment specialization with worst-case cross-environment robustness.

Let $R_k^*$ denote the Bayes risk for environment $k$:

$$R_k^* = \min_{f:\mathcal{X}\to\mathcal{Y}} \mathbb{E}_{(\mathbf{x},y)\sim\mathcal{P}_k}\left[\ell(f(\mathbf{x}), y)\right] \tag{29}$$

This minimum is achieved by the Bayes optimal predictor $f_k^*$ for environment $k$. Let $\theta^* = \{\phi^*, \omega_1^*, \ldots, \omega_m^*\}$ be a set of parameters that globally minimizes $L(\theta)$. We define the deviation from native Bayes risk for expert $k$ under $\theta^*$ as $\delta_k(\theta^*) = \mathcal{R}_{k,k}(\theta^*) - R_k^*$. Note that $\delta_k(\theta^*) \geq 0$.

We consider a hypothetical parameter configuration, denoted $\theta_{Bayes} = \{\phi_{Bayes}, \omega_{1,Bayes}, \ldots, \omega_{m,Bayes}\}$, where each expert achieves its native Bayes risk. That is, $\mathcal{R}_{k,k}(\theta_{Bayes}) = R_k^*$ for all $k \in \{1, \ldots, m\}$. The existence of such a $\theta_{Bayes}$ within the model's hypothesis space is assumed under conditions of sufficient model capacity. Let $M(\theta) = \max_{i,j} \mathcal{R}_{i,j}(\theta)$. We can then define $M_{Bayes} = M(\theta_{Bayes})$ and $M^* = M(\theta^*)$.

## D.2   Bound on deviation from native Bayes optimality

The MEDRO objective, while promoting robustness, also includes a specialization term $\sum_{k=1}^m \mathcal{R}_{k,k}(\theta)$ that encourages each expert $\omega_k \circ \phi$ to perform well on its native environment $\mathcal{P}_k$. The following proposition quantifies how far the native risks $\mathcal{R}_{k,k}(\theta^*)$ might deviate from their theoretical minima $R_k^*$ at the MEDRO optimum $\theta^*$.

**Proposition 1 (bound on deviation from native Bayes optimality)** *Let $\theta^*$ be a global minimizer of the MEDRO objective $L(\theta)$ as defined in Eq. (28), with $\gamma > 0$. Under the assumption that a configuration $\theta_{Bayes}$ exists such that $\mathcal{R}_{k,k}(\theta_{Bayes}) = R_k^*$ for all $k$, the sum of deviations from native Bayes risks at $\theta^*$ is bounded as:*

$$\sum_{k=1}^{m} \delta_k(\theta^*) \leq \gamma \left( M_{Bayes} - M^* \right) \tag{30}$$

*Furthermore, since $M^* \geq 0$ (assuming non-negative risks), a simpler bound is:*

$$\sum_{k=1}^{m} \delta_k(\theta^*) \leq \gamma M_{Bayes} \tag{31}$$

*Proof.* By the optimality of $\theta^*$, we have $L(\theta^*) \leq L(\theta_{Bayes})$. Substituting the definitions:

$$\sum_{k=1}^{m} \mathcal{R}_{k,k}(\theta^*) + \gamma M^* \leq \sum_{k=1}^{m} \mathcal{R}_{k,k}(\theta_{Bayes}) + \gamma M_{Bayes}$$

Using $\mathcal{R}_{k,k}(\theta^*) = R_k^* + \delta_k(\theta^*)$ and $\mathcal{R}_{k,k}(\theta_{Bayes}) = R_k^*$:

$$\sum_{k=1}^{m} (R_k^* + \delta_k(\theta^*)) + \gamma M^* \leq \sum_{k=1}^{m} R_k^* + \gamma M_{Bayes}$$

The $\sum R_k^*$ terms cancel:

$$\sum_{k=1}^{m} \delta_k(\theta^*) + \gamma M^* \leq \gamma M_{Bayes}$$

Rearranging yields the first bound in Eq. (30). Since $M_{Bayes} - M^* \leq M_{Bayes}$ (as $M^* \geq 0$), the second bound in Eq. (31) also holds. $\square$

**Discussion** Proposition 1 demonstrates that the total deviation from native Bayes optimality, $\sum \delta_k(\theta^*)$, is controlled by $\gamma$ and the extent of robustness improvement, $M_{Bayes} - M^*$. The deviation $\sum \delta_k(\theta^*)$ will be small if (i) $\gamma$ is chosen to be sufficiently small; or (ii) the term $(M_{Bayes} - M^*)$ is small. This latter case occurs if $M_{Bayes}$ itself is small (i.e., native Bayes optimal predictors are inherently robust), or if $M_{Bayes}$ is not significantly larger than $M^*$ (i.e., there's limited scope for MEDRO to improve worst-case robustness beyond what $\theta_{Bayes}$ already offers, possibly in "robustness-hard" problem settings where further reduction in $M^*$ incurs substantial costs to native performance).

## D.3 Assumption on expert optimality

The specialization term $\sum_{k=1}^{m} \mathcal{R}_{k,k}(\theta)$ in the MEDRO objective directly encourages each expert $\omega_k \circ \phi$ to minimize risk on its corresponding environment $\mathcal{P}_k$. Proposition 1 provides a quantitative framework for understanding when the resulting native risks $\mathcal{R}_{k,k}(\theta^*)$ will be close to their theoretical minima $R_k^*$. For the subsequent analysis concerning the properties of the learned representation $\phi^*$, we make the following assumption:

**Assumption 1 (expert optimality)** *For each environment $i \in \{1, \ldots, m\}$, the expert head $\omega_i^*$ combined with the shared feature extractor $\phi^*$ at the MEDRO optimum $\theta^*$ achieves, or closely approximates, the Bayes-optimal prediction on environment $i$:*

$$\omega_i^* \circ \phi^* \approx \arg \min_{f:\mathcal{X} \to \mathcal{Y}} \mathbb{E}_{(\mathbf{x},y) \sim \mathcal{P}_i} \left[ \ell(f(\mathbf{x}), y) \right] \tag{32}$$

*This is equivalent to assuming that each deviation $\delta_i(\theta^*)$ is small or negligible.*

**Justification** This assumption is grounded in the structure of the MEDRO objective and the insights from Proposition 1. It posits that under ideal conditions (e.g., sufficient model capacity, successful convergence to a global optimum $\theta^*$) and an appropriate choice of $\gamma$, the balance struck by the MEDRO objective is such that the conditions for the bound in Proposition 1 to be small (e.g., small $\gamma$, or a problem structure where $M_{Bayes} - M^*$ is small for reasons not presupposing advanced feature invariance) are effectively met. This ensures that experts remain highly proficient in their native domains.

## D.4 Domain-invariant conditionals under equalized risks

Building upon the assumption of expert optimality, we now explore the properties of the learned representation $\phi^*$ under a particularly strong and idealized scenario of cross-environment generalization. We consider a situation where not only does each expert $\omega_i^*$ achieve near-Bayes optimality on its native domain $\mathcal{P}_i$ (per Assumption 1), but also its performance level on any other domain $\mathcal{P}_j$ matches this optimal native performance. This implies that $\mathcal{R}_{i,j}(\theta^*) \approx \mathcal{R}_{i,i}(\theta^*) \approx R_i^*$ for all $j$.

While the MEDRO objective aims to minimize the single worst-case risk $\max_{a,b} \mathcal{R}_{a,b}(\theta)$, it does not explicitly enforce that all $\mathcal{R}_{i,j}$ are reduced precisely to $R_i^*$. However, if the minimization of $\max_{a,b} \mathcal{R}_{a,b}$ were so effective that all such risks were driven down to this fundamental limit, the following proposition elucidates a crucial property of $\phi^*$. Proposition 2 thus explores the structure of $\phi^*$ in such an idealized state of uniform expert generalization.

**Proposition 2 (domain-invariant conditionals)** *Under Assumption 1 (Expert Optimality), if an MEDRO solution $\theta^* = \{\phi^*, \omega_1^*, \ldots, \omega_m^*\}$ achieves equal risk across all source domain pairs, i.e., for all $i, j \in \{1, \ldots, m\}$,*

$$\mathcal{R}_{i,j}(\theta^*) = \mathcal{R}_{i,i}(\theta^*) \tag{33}$$

*then the conditional distribution of labels given the learned representation $\phi^*(\mathbf{x})$ is invariant across these domains:*

$$\mathcal{P}_i\big(Y \mid \phi^*(\mathbf{x})\big) = \mathcal{P}_j\big(Y \mid \phi^*(\mathbf{x})\big) \quad \forall i, j \in \{1, \ldots, m\}, \text{ for } \phi^*(\mathbf{x}) \text{ in the support.} \tag{34}$$

*Proof.* By Assumption 1, $\omega_i^* \circ \phi^*$ (denoted $h_i(\phi^*(\mathbf{x}))$ for brevity) achieves (near) Bayes-optimal risk on environment $i$. Thus, $\mathcal{R}_{i,i}(\theta^*) \approx R_i^*$, the minimal possible risk on environment $i$ given the representation $\phi^*(\mathbf{x})$. The condition in Eq. (33), $\mathcal{R}_{i,j}(\theta^*) = \mathcal{R}_{i,i}(\theta^*)$, implies that $h_i(\phi^*(\mathbf{x}))$ also achieves this same (minimal) risk $R_i^*$ when evaluated on data from environment $j$. This means $h_i(\phi^*(\mathbf{x}))$ is effectively Bayes-optimal for predicting $Y$ from $\phi^*(\mathbf{x})$ under $\mathcal{P}_j$ as well, and achieves the same risk value $R_i^*$.

Let $\mathbf{z} = \phi^*(\mathbf{x})$. The Bayes-optimal predictor for environment $k$ using representation $\mathbf{z}$ is $f_k^*(\mathbf{z}) = \arg\min_{y' \in \mathcal{Y}} \mathbb{E}_{Y \sim \mathcal{P}_k(Y|\mathbf{Z}=\mathbf{z})}[\ell(y', Y)]$. Our premise is that $\omega_i^*(\mathbf{z})$ serves as $f_i^*(\mathbf{z})$ (achieving risk $R_i^*$) and also as $f_j^*(\mathbf{z})$ (achieving the same risk $R_i^*$). For the same predictor $\omega_i^*(\cdot)$ to be Bayes-optimal for two different conditional distributions $\mathcal{P}_i(Y|\mathbf{Z} = \mathbf{z})$ and $\mathcal{P}_j(Y|\mathbf{Z} = \mathbf{z})$ and yield the same Bayes risk value, these conditional distributions must be identical. If they were different, say $\mathcal{P}_i(Y|\mathbf{z}) \neq \mathcal{P}_j(Y|\mathbf{z})$ for some $\mathbf{z}$ in the support, then $f_i^*(\mathbf{z})$ would generally differ from $f_j^*(\mathbf{z})$, or if they were the same function, the achieved Bayes risks would generally differ, contradicting the premise that $\omega_i^*(\mathbf{z})$ achieves risk $R_i^*$ for both. Thus, $\mathcal{P}_i(Y|\phi^*(\mathbf{x}) = \mathbf{z}) = \mathcal{P}_j(Y|\phi^*(\mathbf{x}) = \mathbf{z})$ for all $\mathbf{z}$ in the common support of the representations across these domains. $\square$

**Remark (approximate domain invariance)** If for all $i, j$, $\big|\mathcal{R}_{i,j}(\theta^*) - \mathcal{R}_{i,i}(\theta^*)\big| \leq \varepsilon$, then under mild smoothness assumptions on the loss, the conditional distributions $\mathcal{P}_i(Y \mid \phi^*(x))$ and $\mathcal{P}_j(Y \mid \phi^*(x))$ differ by at most $O(\sqrt{\varepsilon})$ in total variation.

**Implication** Proposition 2 suggests that if MEDRO can find a solution where each expert is not only optimal in its native domain but also performs identically well (at its optimal native-domain risk level) across all other source domains, then the learned feature extractor $\phi^*$ must capture a representation that neutralizes domain-specific variations relevant to $P(Y|\mathbf{X})$, revealing a core, domain-invariant conditional relationship $P(Y|\phi^*(\mathbf{X}))$. This is a highly desirable property for out-of-distribution generalization.

# E Experimental details and hyperparameter search

Our codes are available at https://github.com/jyjeongku/MEDRO.

## E.1 Controlled experiments on CelebA and Waterbirds

We followed the experimental protocol introduced in the GroupDRO paper [6] to evaluate MEDRO under controlled binary classification tasks. These experiments were conducted on the CelebA and

Table 5: Summary of benchmark datasets used in our experiments.

| Dataset | Modality | # Samples | # Classes | Domains/Attributes | Task | Shift Type |
|---|---|---|---|---|---|---|
| Waterbirds | Natural image | 11,788 | 2 | 2 (background) | Bird classification | Subpop. |
| CelebA | Natural image | 202,599 | 2 | 2 (gender) | Hair color prediction | Subpop. |
| CivilComments | Text | 448,000 | 2 | 8 (demographics) | Toxicity detection | Subpop. |
| MultiNLI | Text | 392,702 | 3 | 2 (negation) | Natural language inference | Subpop. |
| MetaShift | Natural image | 3,499 | 2 | 2 (background) | Cat/Dog classification | Subpop. |
| NICO++ | Natural image | 88,866 | 60 | 6 (background) | Multi-category classification | Subpop. |
| CheXpert | Biomedical image | 222,792 | 2 | 6 (race×gender) | Disease diagnosis | Subpop. |
| Living17 | Natural image | 45,900 | 17 | N/A (Attr. generalization) | Multi-category classification | Subpop. |
| PovertyMap | Satellite image | 19,669 | Regression | 23×2 (country/area) | Asset wealth prediction | Hybrid |
| Camelyon17 | Biomedical image | 455,954 | 2 | 5 (hospitals) | Tumor detection | Domain |
| iWildCam | Natural image | 203,029 | 182 | 323 (locations) | Animal classification | Domain |

Waterbirds datasets, which exhibit clear subpopulation shifts due to strong correlations between target labels and spurious attributes. We compared MEDRO against ERM and GroupDRO under the same backbone architecture and optimization settings, using known group labels for both training and validation.

### E.1.1   CelebA

We used the binary attribute `Blond_Hair` as the prediction target and `Male` as the confounding variable. This results in four subgroups based on combinations of hair color and gender. The training set is highly imbalanced, with the smallest group (blond-haired males) accounting for only a small fraction of samples.

**Training subgroup sizes:**

- (0, 0): 71,629    (not blond, female)
- (0, 1): 66,874    (not blond, male)
- (1, 0): 22,880    (blond, female)
- (1, 1): 1,387    (blond, male)

Models were trained for 50 epochs using a pretrained ResNet-50 backbone [51]. We used SGD with momentum 0.9, batch size 128, learning rate $10^{-5}$, and strong $\ell_2$ regularization (weight decay = 0.1). No data augmentation was used.

We evaluated all methods using the final epoch model (i.e., without validation-based model selection), as GroupDRO typically converges near the end of training and our setup reused its original fixed hyperparameters. A 10% validation split was retained for consistency with GroupDRO, but it was not used for early stopping or model selection.

In both GroupDRO and MEDRO, we followed the same robust optimization setup by setting the step size for the adversarial group weighting update (also called DRO step size) to 0.01, as used in the original GroupDRO implementation.

The balance factor $\gamma$ was selected as 0.5 based on sensitivity analysis conducted on both CelebA and Waterbirds under strong $\ell_2$ regularization. We reported results averaged over five runs with different random seeds.

### E.1.2   Waterbirds

We followed the same setup for the Waterbirds dataset (also known as CUB), where the binary target label `waterbird_complete95` indicates whether the bird is a waterbird (1) or landbird (0), and the confounder `forest2water2` denotes the background environment (water or forest). This setup produces four subgroups with strong spurious correlations.

**Training subgroup sizes:**

- (0, 0): 3,498    (landbird on land)
- (0, 1): 184    (landbird on water)
- (1, 0): 56    (waterbird on land)

- (1, 1): 1,057    (waterbird on water)

Models were trained for 300 epochs using the same ResNet-50 backbone. We used SGD (momentum 0.9), batch size 128, learning rate $10^{-5}$, and weight decay 1.0. No data augmentation was used.

We again evaluated using the final epoch model, without validation-based model selection. The 10% validation split was retained for comparability but unused in selection.

As in CelebA, we set the DRO step size to 0.01, consistent with the GroupDRO implementation. The same value $\gamma = 0.5$ was used for MEDRO, as determined from sensitivity analysis. No additional tuning was conducted for Waterbirds.

## E.2 SubpopBench settings

### E.2.1 Benchmark overview

SubpopBench covers a wide range of real-world datasets and subpopulation structures. We focused on eight representative tasks—Waterbirds, CelebA, CivilComments, MultiNLI, MetaShift, NICO++, CheXpert, and Living17—chosen to span different modalities and types of subpopulation shift. For all datasets, we followed the SubpopBench-provided data splits, group definitions, and preprocessing procedures without modification.

### E.2.2 Evaluation metric

Consistent with the official protocol, we adopted worst-group accuracy as the primary evaluation metric. It is the standard criterion used in the benchmark and enables direct comparison across methods under a shared notion of robustness. Although it does not capture all performance trade-offs (e.g., precision-recall balance), worst-group accuracy remains the most widely accepted measure of subgroup-level reliability, particularly in scenarios where minimizing failure on the most vulnerable group is critical.

### E.2.3 Model selection

We adopted the oracle selection setting, where group attributes were available during both training and validation. This is considered the most ideal scenario in the benchmark specification, allowing test set worst-group accuracy to identify optimal algorithm performance. While not feasible in real-world deployment, this setup provides a standardized estimate of each method's potential under full group supervision. It enables fair comparison by isolating algorithmic differences from model selection challenges.

### E.2.4 Hyperparameter search

For MEDRO, we followed the official protocol, tuning across 16 randomized hyperparameter configurations. Each configuration was drawn from a predefined search space in Table 6. Best configurations were selected based on validation worst-group accuracy, using group labels as specified in the protocol. For final evaluation, MEDRO was retrained using the best hyperparameters and evaluated with three different random seeds.

To ensure consistency across tasks, we fixed the balance factor $\gamma$ to 1 in all experiments. This choice simplifies tuning and reflects a realistic scenario for evaluating MEDRO's generalization under a uniform configuration.

In addition to tuning the model learning rate $\eta_\theta$ (as part of the general search space), we tuned the risk weight step size $\eta_\Lambda$, which controls the group weight update during training. The range $10^{\mathrm{Uniform}[-3,-1]}$ was used, consistent with the GroupDRO setting.

### E.2.5 Implementation details

We followed the SubpopBench protocol for backbone, preprocessing, and optimization. For image datasets, we used ResNet-50 pretrained on ImageNet-1K [51]; for text datasets, we used BERT-base (bert-base-uncased) [52]. Image inputs were resized and center-cropped to 224×224, then normalized using ImageNet statistics. Image models were trained with SGD (momentum 0.9), and text models

Table 6: Search space for general hyperparameters used for MEDRO in our SubpopBench experiments. All values were sampled per run via random search.

| Backbone | Parameter | Search Range |
|---|---|---|
| ResNet-50 | Learning rate | $10^{\mathrm{Uniform}[-4,-2]}$ |
| | Weight decay | $10^{\mathrm{Uniform}[-6,-3]}$ |
| | Batch size | $2^{\mathrm{Uniform}[6,6.75]}$ |
| | Dropout | 0.0 (fixed) |
| BERT-base | Learning rate | $10^{\mathrm{Uniform}[-5.5,-4]}$ |
| | Weight decay | $10^{\mathrm{Uniform}[-6,-3]}$ |
| | Batch size | $2^{\mathrm{Uniform}[3,5.5]}$ |
| | Dropout | Choice$\{0.0, 0.1, 0.5\}$ |

with AdamW [53]. Training steps followed the SubpopBench standard: 5k for Waterbirds and MetaShift, 20k for CheXpert, 30k for CelebA, CivilComments, MultiNLI, and NICO++, and 60k for Living17.

### E.2.6 Isolating ensemble effects: Multi-head architecture analysis

To determine whether MEDRO's performance improvements stem from its expanded uncertainty formulation or from ensemble effects of using multiple expert heads, we conducted a controlled comparison with GroupDRO applied to a multi-head architecture. We implemented this variant with the following characteristics:

- Same multi-head structure ($m$ heads + shared feature extractor)
- Each head independently performs GroupDRO optimization
- Total loss = average of GroupDRO losses across heads
- Same ensemble inference at test time

Heads differ only in random initialization and are not environment-specific experts. We evaluated this variant on eight SubpopBench datasets using the same hyperparameter search protocol as other baselines.

Table 7 presents the comparison. GroupDRO with multi-head architecture achieves 67.8% overall worst-group accuracy, a +0.4% difference from single-head GroupDRO (67.4%). MEDRO achieves 69.5%, a +1.7% difference from the multi-head variant. These results indicate that the performance difference between MEDRO and single-head GroupDRO (+2.1%) exceeds the difference between multi-head and single-head GroupDRO (+0.4%). This suggests that MEDRO's gains arise primarily from its algorithmic formulation rather than from ensemble effects.

Table 7: Worst-group accuracy (%) comparison across SubpopBench datasets. Mean and standard deviation over three runs.

| Method | Waterbirds | CelebA | CivilC. | MultiNLI | MetaShift | NICO++ | CheXpert | Living17 | Overall |
|---|---|---|---|---|---|---|---|---|---|
| GroupDRO | 78.6 ±1.0 | 89.0 ±0.7 | 70.6 ±1.2 | 76.0 ±0.7 | 85.6 ±0.4 | 37.8 ±1.8 | 74.5 ±0.2 | 27.2 ±1.5 | 67.4 |
| GroupDRO (w/ multi-head) | 81.1 ±1.3 | 90.2 ±0.3 | 71.5 ±1.4 | 74.8 ±1.3 | 85.4 ±1.0 | 36.7 ±2.4 | 73.5 ±0.1 | 28.8 ±1.1 | 67.8 |
| **MEDRO** | 83.8 ±1.3 | 90.4 ±0.4 | 73.4 ±0.5 | 75.7 ±0.8 | 85.9 ±0.4 | 39.1 ±2.5 | 75.0 ±0.4 | 32.6 ±0.7 | 69.5 |

### E.2.7 Baseline method descriptions

We compared MEDRO against 19 baseline methods as implemented and defined in SubpopBench [12]. These span a broad spectrum of approaches for mitigating subpopulation shift, including robust optimization, data augmentation, loss reweighting, and two-stage learning:

- **ERM** [31]: Standard empirical risk minimization without robustness interventions.

- **GroupDRO** [6]: Minimizes worst-group risk by dynamically upweighting poorly performing groups during training.

- **IRM** [13]: Encourages predictors to remain invariant across multiple environments by enforcing equal optimality of predictors across domains.

- **Mixup** [54]: Regularizes training by interpolating random pairs of inputs and labels to generate synthetic examples.

- **JTT** [55]: Two-stage method that identifies high-loss examples via ERM and retrains a model by upsampling them once.

- **DFR** [56], **CRT** [57], **ReWeightCRT**: Two-stage methods that first learn representations via ERM and then retrain the classifier head on a balanced or reweighted dataset.

- **LfF** [58]: Simultaneously trains a biased model and a main model, reweighting samples in the latter based on difficulty estimated by the former.

- **LISA** [21]: Uses selective mixup across domains and classes to promote invariant prediction while reducing reliance on spurious features.

- **CVaRDRO** [59]: Minimizes conditional value-at-risk over per-group losses to target high-loss groups.

- **MMD** [60]: Minimizes the maximum mean discrepancy between group feature distributions to align representations.

- **CBLoss** [61], **Focal** [62], **LDAM** [63]: Loss-level modifications designed to address class imbalance: CBLoss and Focal adjust per-sample loss weights, while LDAM adds class-dependent margins to logits improve separation.

- **ReSample, ReWeight, SqrtReWeight** [64]: Sampling- or weighting-based methods that rebalance subgroup proportions during training.

- **BSoftmax** [65]: Adjusts softmax normalization to account for class imbalance by using empirical class frequencies.

### E.3  WILDS settings

We evaluated MEDRO on three domain generalization tasks from the WILDS benchmark: Camelyon17, iWildCam, PovertyMap. All experiments followed the official WILDS codebase and evaluation protocol without modification. The balance factor $\gamma$ was fixed to 1 for all tasks, consistent with our SubpopBench configuration.

Table 8 summarizes the training configurations used for MEDRO on WILDS datasets. For Camelyon17, we enabled ImageNet pretraining, following prior work [18] that demonstrated improved performance with this setting.

Following the original GroupDRO implementation, we used a default risk weight step size $\eta_\Lambda = 0.01$ for MEDRO. For iWildCam, we found that a smaller step size of $\eta_\Lambda = 0.0001$ improved training stability and therefore adopted this value. We also enabled data augmentation specifically for iWildCam, using the corresponding option in the WILDS codebase, as it was found to enhance generalization in this setting.

#### E.3.1  Baseline method descriptions

We compared MEDRO against 11 baseline methods commonly used in domain generalization, as evaluated in the WILDS benchmark. These methods span diverse strategies such as invariant representation learning, adversarial alignment, risk variance minimization, data augmentation, model averaging, and two-stage selection correction:

- **ERM** [31]: Standard empirical risk minimization over pooled domains, without explicit robustness to domain shift.

- **GroupDRO** [6]: Minimizes the worst-case domain risk by upweighting domains with higher current loss during training.

- **IRM** [13]: Learns representations such that a shared optimal classifier performs well across all training domains.

Table 8: Training configurations used for MEDRO on the WILDS datasets. Learning rate and weight decay were selected from the official WILDS search grids. For PovertyMap, the learning rate decays by a factor of 0.96 per epoch. Other settings follow the benchmark defaults unless otherwise noted.

| Parameter | Camelyon17 | iWildCam | PovertyMap |
|---|---|---|---|
| Backbone | DenseNet-121 | ResNet-50 | ResNet-18-MS |
| ImageNet Pretrained | True | True | True |
| Data Augmentation | None | RandAugment | None |
| Optimizer | SGD | Adam | Adam |
| Learning Rate | $10^{-4}$ | $3 \times 10^{-5}$ | $10^{-3}$ |
| Weight Decay | $10^{-2}$ | 0 | 0 |
| Batch Size | 32 | 16 | 64 |
| Epochs | 10 | 12 | 200 |

- **CORAL** [16]: Aligns feature distributions across domains by minimizing discrepancies in second-order statistics (mean and covariance).

- **DANN** [15]: Uses adversarial training to learn domain-invariant features by confusing a domain classifier.

- **VREx** [9]: Minimizes the variance in per-domain training risks to enforce uniform performance across domains.

- **LISA** [21]: Applies selective mixup augmentation between domains and classes to weaken spurious correlations and promote invariant prediction.

- **Fish** [23]: Encourages gradient alignment across domains to learn representations that generalize consistently.

- **SWAD** [66]: Averages model weights during training to find flatter minima that generalize better to unseen domains.

- **L2A-OT** [22]: Synthesizes pseudo-novel domains via optimal transport-based data augmentation to expand domain diversity.

- **HeckmanDG** [18]: Models domain selection bias via a two-stage Heckman-style estimator to correct for distributional mismatch during training.

