# OpenReview forum: "Multi-Expert Distributionally Robust Optimization for Out-of-Distribution Generalization"
_NeurIPS.cc/2025/Conference — NeurIPS 2025 poster_

### Official Review · Reviewer_J412 · 2025-07-02

**Clarity:** 3
**Significance:** 2
**Originality:** 2
**Rating:** 4
**Confidence:** 3

**Summary:**

In this paper, the authors propose a distributionally robust optimization (DRO) model, MEDRO, extending the well-known GroupDRO by incorporating a multi-experts learning architecture. The idea is that by introducing the extra expert-environment interactions, more spurious relations can be identified and eliminated via optimization. The authors justify the methodology via thorough theoretical studies. The effectiveness of MEDRO over prior works is then confirmed via comprehensive experiments.

**Questions:**

1. About the guideline of choosing ensemble or gating function MEDRO in line 197-199, does the validation data follow train or test data distribution?
2. Is MEDRO (ours) in Table 1 based on simple ensemble or gating function? Also, do the authors consider reporting performance of both ensemble and gating function for all the experiments?
3. Why are the ERM’s performance on CelebA and Waterbirds significantly different between Table 2 and 3?
4. About Figure 2, maybe aligning the y-axes of the two plots, so that the impacts on average and worst-case performance can be compared easily?

**Ethical Concerns:**

["NO or VERY MINOR ethics concerns only"]

**Final Justification:**

The authors resolve the concern that the perfomance comes from ensemble instead of the desired cross-environment optimization. The new experiments suggest the advantage from the core algorithm is solid. Also, the authors help clarify the experiment settings, justifying the evaluation results. The overall quality of this work is good, while the novelty of the methodology is slightly limited. I therefore recommend borderline accept.

**Limitations:**

yes

**Paper Formatting Concerns:**

No concern

**Quality:**

3

**Strengths And Weaknesses:**

**Strengths**
1. The authors build a solid theoretical foundation supporting the arguments and the proposed model.
2. The experiments are comprehensive in terms of both distribution shifts and out-of-distribution generalization methods.
3. The writing is impressive. The authors do a great job presenting the background knowledge and their research under the page limit.

**Weaknesses**
1. Whether the performance gain comes from ensemble may be unclear. MEDRO natively possesses the ensemble mechanism but most of the other models do not.Therefore, for example, can we expect the performance gap becomes smaller if we implement GroupDRO with ensemble?
2. Although the proposed algorithm is reasonable, it is not particularly surprising.
3. Some details of experiments are unclear and may impact the justification of model performance. ( See Question 1 and 2)

---

> ### Author Rebuttal · Authors · 2025-07-31
>
> ### `Weakness 1:`
> > "Whether the performance gain comes from ensemble may be unclear. MEDRO natively possesses the ensemble mechanism but most of the other models do not. Therefore, for example, can we expect the performance gap becomes smaller if we implement GroupDRO with ensemble?"
>
> Thank you for raising this important point about disentangling the ensemble effect from MEDRO's core contribution. To address this concern, we implemented Multi-Head GroupDRO with the same architecture as MEDRO:
> - Same multi-head structure ($m$ expert heads + shared feature extractor)
> - Each head independently performs GroupDRO optimization
> - Total loss = average of $m$ GroupDRO losses
> - Same ensemble inference at test time
>
> Note that Multi-head GroupDRO's heads are not environment-specific experts, but merely differ in random initialization---each head minimizes the same robust objective independently.
>
> Results demonstrate clear separation between ensemble effects and MEDRO's core contribution.
>
> Key findings:
> 1. Multi-Head GroupDRO shows minimal improvement over single-head GroupDRO (+0.2% on CelebA, +0.8% on Waterbirds), confirming limited ensemble benefits in this setting.
> 2. MEDRO significantly outperforms Multi-Head GroupDRO (+3.5% on CelebA, +1.4% on Waterbirds), demonstrating that our gains primarily come from modeling cross-environment risks.
> 3. SubpopBench results further validate this: across 6 datasets, Multi-Head GroupDRO achieves 79.4% (only +0.3% over single-head), while MEDRO achieves 80.7% (+1.3% over multi-head).
> 4. This experiment clearly shows that MEDRO's advantage stems from its expanded uncertainty set created by modeling cross-environment risks, not merely the ensemble mechanism.
>
> The performance gap remains significant even after controlling for ensemble effects, validating that our method's contribution goes beyond architectural choices to a fundamentally different optimization objective.
> | | CelebA | | Waterbirds | |
> |--------|--------|-------|-----------|-------|
> | **Method** | **Avg.** | **Worst** | **Avg.** | **Worst** |
> | GroupDRO | 93.7 (±0.3) | 85.7 (±2.2) | 96.1 (±1.3) | 84.1 (±2.4) |
> | GroupDRO (w/ multi-head) | 93.3 (±0.4) | 85.9 (±1.4) | 93.7 (±1.2) | 84.9 (±1.6) |
> | MEDRO | 92.2 (±0.8) | 89.4 (±1.2) | 94.2 (±1.8) | 86.3 (±1.9) |
>
> | Method | Waterbirds | CelebA | CivilC. | MultiNLI | MetaShift | CheXpert | Overall |
> |--------|------------|--------|---------|----------|-----------|----------|---------|
> | GroupDRO | 78.6 (±1.0) | 89.0 (±0.7) | 70.6 (±1.2) | **76.0** (±0.7) | 85.6 (±0.4) | 74.5 (±0.2) | 79.1 |
> | GroupDRO (w/ multi-head) | 81.1 (±1.3) | 90.2 (±0.3) | 71.5 (±1.4) | 74.8 (±1.3) | 85.4 (±1.0) | 73.5 (±0.1) | 79.4 |
> | **MEDRO** | **83.8** (±1.3) | **90.4** (±0.4) | **73.4** (±0.5) | 75.7 (±0.8) | **85.9** (±0.4) | **75.0** (±0.4) | **80.7** |
>
> ### `Weakness 2:`
> >"Although the proposed algorithm is reasonable, it is not particularly surprising."
>
> Thank you for this thoughtful comment. While we understand why MEDRO might appear as a straightforward extension of GroupDRO, we respectfully argue that it addresses fundamental limitations through a principled new approach.
>
> **Cross-environment risk modeling**
>
> The key difference lies in what we optimize. GroupDRO minimizes the worst-case risk over m environments using a single classifier. MEDRO introduces cross-environment risks R_{i,j}—the risk when expert i processes data from environment j—creating m² distinct risks to optimize. This is not merely quantitative expansion. Each R_{i,j} captures a specific failure mode. For instance, in Waterbirds, R_{land, water} measures how an expert trained on land background fails when processing water backgrounds. By explicitly minimizing all such mismatches, MEDRO pressures the shared feature extractor φ to learn background-invariant representations.
>
> **Overcoming the weighted ERM limitation**
>
> Critically, GroupDRO as weighted ERM has fundamental limitations in learning invariant features (IRM, Proposition 2). It can only reweight samples, not change the representation itself.
>
> MEDRO's architecture—multiple **expert** heads with a shared feature extractor—combined with our objective that balances specialization (Σᵢ R_{i,i}) and cross-environment robustness (max_{i,j} R_{i,j}) encourages invariant representation learning. Our analysis (Appendix D) shows that when cross-environment risks are equalized (R_{i,j} ≈ R_{i,i}), the learned features tend toward satisfying P_i(Y|φ*(x)) ≈ P_j(Y|φ*(x)). This theoretical connection emerges naturally from our objective.
>
> Importantly, optimizing m² risks does not significantly slow convergence: we maintain O(1/√T) rate with only a logarithmic factor difference (Appendix C).
>
> In summary, MEDRO's contribution lies in recognizing that different expert-environment interactions represent distinct failure modes worth modeling explicitly, and showing how this leads to both theoretical insights and empirical improvements.
>
> ### `Weakness 3:`
> >"Some details of experiments are unclear and may impact the justification of model performance. (See Question 1 and 2)"
>
> We appreciate you pointing out these important experimental details. As addressed in our responses to Questions 1 and 2:
>
> - **Validation distributions**: We have clarified that validation sets follow different distribution patterns depending on the task type (ID for domain generalization, same groups but different proportions for subpopulation shift).
>
> - **Inference strategy**: We have provided complete results for both strategies. We reported selectively based on context: ensemble for fair comparison with single-stage methods (Table 1), both when two-stage methods exist (Table 2), and ensemble for domain shift tasks where gating offers limited OOD benefits (Table 3-4).
>
> These clarifications demonstrate that our experimental choices were principled and consistent. We will revise the manuscript to make these methodological details explicit upfront, ensuring full transparency.
>
> ### `Question 1:`
> >"About the guideline of choosing ensemble or gating function MEDRO in line 197-199, does the validation data follow train or test data distribution?"
>
> Thank you for this clarification question. The validation distributions are:
>
> - **Domain Generalization**: ID validation set (follows train distribution, while test is out-of distribution)
>
> - **Subpopulation shift**: All datasets (train/valid/test) contain the same groups, but the group proportions typically differ across splits
>
> We will incorporate these clarifications into the revised manuscript to ensure readers clearly understand our experimental setup and choices.
>
> ### `Question 2:`
> >"Is MEDRO (ours) in Table 1 based on simple ensemble or gating function? Also, do the authors consider reporting performance of both ensemble and gating function for all the experiments?"
>
> Thank you for this important clarification question. The MEDRO results in Table 1 are based on the simple ensemble approach.
>
> We actually evaluated both inference strategies across all experiments during our research. However, we selectively reported results based on the experimental context:
>
> - **Table 1 (GroupDRO benchmark)**: We reported only simple ensemble to ensure fair comparison with 1-stage baselines (ERM, GroupDRO)
> - **Table 2 (SubpopBench)**: We included both strategies since the benchmark already contains 2-stage methods like DFR, CRT, and JTT
> - **Table 3 & 4 (WILDS)**: We focused on simple ensemble as gating networks trained on ID validation data provide limited benefits for OOD test performance
>
> Per your request, here are the complete results for both inference strategies across all experiments:
> | | CelebA | | Waterbirds | |
> |--------|--------|-------|-----------|-------|
> | **Method** | **Avg.** | **Worst** | **Avg.** | **Worst** |
> | MEDRO | 92.2 (±0.8) | 89.4 (±1.2) | 94.2 (±1.8) | 86.3 (±1.9) |
> | MEDRO (w/ gating) | 92.3 (±0.5) | 90.1 (±0.8) | 92.6 (±1.7) | 88.7 (±2.2) |
>
> | | Camelyon17 | | iWildCam | |
> |--------|--------|-------|-----------|-------|
> | **Method** | **Avg.** | **Worst** | **Avg.** | **Worst** |
> | MEDRO | 92.6 (±0.5) | 87.8 (±1.9) | 34.1 (±0.7) | 31.5 (±0.8) |
> | MEDRO (w/ gating) | 92.7 (±0.4) | 87.8 (±2.0) | 36.4 (±1.1) | 31.6 (±0.8) |
>
> **PovertyMap**
> | Method | Avg. Val | Avg. Test | Worst Val | Worst Test |
> |--------|-------------|--------------|-----------------|------------------|
> | MEDRO | 0.80 (±0.04) | 0.80 (±0.03) | 0.50 (±0.06) | 0.49 (±0.03) |
> | MEDRO (w/ gating)| 0.81 (±0.04) | 0.80 (±0.04) | 0.53 (±0.06) | 0.49 (±0.05) |
>
> As shown, gating provides modest improvements for subpopulation shift tasks but limited benefits for domain generalization. We are grateful for this suggestion and will include a comprehensive table in the appendix showing both inference strategies across all experiments.
>
> ### `Question 3:`
> >"Why are the ERM's performance on CelebA and Waterbirds significantly different between Table 2 and 3?"
>
> We believe the reviewer is referring to the difference between Table 1 and 2 (not Table 2 and 3). The performance difference in ERM stems from different experimental protocols:
>
> - **Table 1** uses fixed hyperparameters from the GroupDRO paper
> - **Table 2** uses SubpopBench's hyperparameter search protocol, which searches over predefined ranges and selects the best configuration
>
> As shown in Table 6 in Appendix E, the hyperparameter search in SubpopBench allows ERM to find more optimal configurations for each dataset, explaining the performance improvement.
>
> ### `Question 4:`
> >"About Figure 2, maybe aligning the y-axes of the two plots, so that the impacts on average and worst-case performance can be compared easily?"
>
> We thank the reviewer for this valuable suggestion. Following your recommendation, we aligned the y-axes of both plots (40-100%), which indeed makes the comparison between average and worst-group performance much clearer. We will include this improved figure in the revised manuscript.

---

> > ### Comment · Reviewer_J412 · 2025-08-04
> >
> > I truly appreciate the authors' very detailed and helpful responses. My main concern, performance gain from ensemble instead of the cross-environment learning, has been resolved. I also thank the authors' efforts addressing the potential clarity issues. I will update my score accordingly soon.

---

> > > ### Author Response · Authors · 2025-08-09
> > >
> > > Thank you for your positive response and constructive feedback. We're glad we could clarify the cross-environment learning benefits and address your concerns.

---

### Official Review · Reviewer_WNwh · 2025-07-02

**Clarity:** 3
**Significance:** 3
**Originality:** 2
**Rating:** 5
**Confidence:** 4

**Summary:**

The authors deal with the GroupDRO problem, which uses distributionally robust optimisation over the convex hull of a set of distributions, each corresponding to one of $m$ environments, samples of each compose the dataset. Here, learning amounts to minimising the worst case loss over all environments. The authors extend this work by suggesting a mixture-of-experts formulation, where, keeping a common representation $\phi$, they learn an expert $ \omega_i\circ \phi $ for each environment. They then optimise for a weigted average of both the ERM loss (assuming a MoE model) as well as the worst case of all experts individually, over all combinations of environments. During prediction (where the environments are not known) the full MoE is used. As a MoE for prediction they either use a gating network, or simply an average of all the experts. They offer good experimental coverage of their method.

**Questions:**

1. How would you expect the performance to change (test accuracy) if the experts were increasingly more constrained to be similar?

**Ethical Concerns:**

["NO or VERY MINOR ethics concerns only"]

**Final Justification:**

This is an incremental work, that is, however, worthwhile: it is well written, while it offers interesting and supportive experimental results.

I maintain my original belief that it should be accepted.

Additionally, the authors addressed certain minor concerns during the rebutal.

**Limitations:**

yes

**Quality:**

3

**Strengths And Weaknesses:**

# Strengths
* This is a good idea, which is well motivated, and the exposition is quite neat and clear.
* A decent amount of theory is provided, albeit hidden in the supplementary.
* The experimental design is thorough, CIs have been provided, and established benchmarks have been used.
* The method performs well in appropriate datasets (CelebA which is large scale and Waterbirds which is a medium-scaled one).
* A good intuition is provided, good coverage of Related works, and good mathematical formalisation.
* MEDRO is a strict generalisation of GroupDRO.



# Weaknesses
* The general idea is relatively simple, but seems well-executed.
* The theory is too hidden in the supplementary materia. I would suggest that key claims are mentioned in the main body of the paper, with references to the proofs in the appendix.
* The method provided is not uniformly superior to other SOTA methods, but does perform on par, and seems to broaden the available toolset of GroupDRO.
* Some directions in the related work to more general methods seem potentially useful. For instance, risk extrapolation [1], CKDRO [2], IPM-DROs [3], etc.


# Notes:
I would like to mention that generally increasing the area over which DRO (and, hence, GroupDRO, MEDRO) is performed, does not necessarilly lead to improvements. This can mean loss of specificity or overly pessimistic regressors/classifiers.\
From this perspective, and to compare the effects of your model to its closest cousin, the GroupDRO, it could perhaps be interesting to perform an analysis on whether/when the extended set of distributions used would be of benefit.

In lieu of a theoretical analysis, one could demonstrate the curve of accuracy as one smoothly moves from the GroupDRO setting to that of MEDRO.

For instance, consider a paradigm where the weights of each head $\omega_i$ were constrained to lie within a ball of constantly increading radius, starting from 0 (GroupDRO) to the actual unconstrained one.

# Suggestions
* I would suggest the use of parentheses in eq 5 around the composition: $ (\omega_i\circ \phi)(x) $, to align with standard practices.

* Also, consider opting for the official sources instead of preprint versions, when applicable. For instance:
   * Improving Domain Generalization with Domain Relations\
      $ \to $ ICLR24
   * Distributionally robust neural networks for group shifts: On the importance of regularization for worst-case generalization\
     $ \to $ Distributionally Robust Neural Networks ICLR20
   * The Risks of Invariant Risk Minimization\
     $ \to $ ICLR21

# References
[1] Y. Chen, J. Kim, and J. Anderson, “Distributionally Robust Decision Making Leveraging Conditional Distributions,” in 2022 IEEE 61st Conference on Decision and Control (CDC), Dec. 2022, pp. 5652–5659. doi: 10.1109/CDC51059.2022.9992782.

[2] D. Krueger et al., “Out-of-Distribution Generalization via Risk Extrapolation (REx),” in Proceedings of the 38th International Conference on Machine Learning, PMLR, Jul. 2021, pp. 5815–5826. Accessed: Jul. 02, 2025. [Online]. Available: https://proceedings.mlr.press/v139/krueger21a.html

[3] M. Staib and S. Jegelka, “Distributionally Robust Optimization and Generalization in Kernel Methods,” in Advances in Neural Information Processing Systems, Curran Associates, Inc., 2019. Accessed: Jul. 02, 2025. [Online]. Available: https://proceedings.neurips.cc/paper/2019/hash/1770ae9e1b6bc9f5fd2841f141557ffb-Abstract.html

---

> ### Author Rebuttal · Authors · 2025-07-31
>
> We thank the reviewer for their insightful and positive feedback. We appreciate the thoughtful assessment and will address the suggestions and questions point by point below.
>
> ---
>
> ### `Response to Weakness 1:`
> > "The general idea is relatively simple, but seems well-executed."
>
> Thank you for recognizing the quality of our execution. We view the simplicity of our core idea as a strength—it suggests we've identified a natural solution to a fundamental problem. The fact that this straightforward insight (cross-environment risk modeling) leads to both theoretical guarantees (domain invariance) and consistent empirical improvements across diverse benchmarks demonstrates its effectiveness. We appreciate your positive assessment of how we developed and validated this approach.
>
> ---
>
> ### `Response to Weakness 2:`
> > "The theory is too hidden in the supplementary material. I would suggest that key claims are mentioned in the main body of the paper, with references to the proofs in the appendix."
>
> Thank you for pointing this out. While we described our theoretical results narratively, we agree they should be more prominent. We will revise to highlight key claims (e.g., as propositions or remarks) within the existing narrative flow, ensuring theoretical contributions are clear while maintaining the paper's readability. Detailed proofs will remain in the appendices.
>
> ---
>
> ### `Response to Weakness 3:`
> > "The method provided is not uniformly superior to other SOTA methods, but does perform on par, and seems to broaden the available toolset of GroupDRO."
>
> We appreciate your accurate assessment of our empirical results. You are correct that MEDRO is not uniformly superior across all settings.
>
> Our contribution is not claiming universal dominance, but rather providing a principled extension of GroupDRO that:
> * Consistently improves over GroupDRO itself (e.g., +3.7% on CelebA, +2.2% on Waterbirds worst-group accuracy)
> * Achieves overall best performance on SubpopBench (80.7% average worst-group accuracy across 6 datasets)
> * Remains highly competitive on domain shift tasks while being designed primarily for subpopulation robustness
>
> As you noted, MEDRO broadens the DRO toolset by explicitly modeling cross-environment risks: a perspective that complements existing approaches. The fact that this principled framework achieves competitive performance across diverse settings while providing theoretical insights (domain invariance connection) validates its value as a new tool in the robustness toolkit rather than a universal replacement.
>
> ---
>
> ### `Response to Weakness 4:`
> > "Some directions in the related work to more general methods seem potentially useful. For instance, risk extrapolation [1], CKDRO [2], IPM-DROs [3], etc."
>
> Thank you for these valuable suggestions for strengthening our related work section. We will incorporate discussions of:
> * **risk extrapolation, REx** [1]: this is indeed a highly relevant work. REx minimizes risk variance across environments to encourage invariance, while MEDRO takes a different approach by explicitly modeling cross-environment risks $\mathcal{R}_{i,j}$. Both methods aim to learn robust representations beyond standard GroupDRO, but through complementary mechanisms.
> * **CKDRO** [2]: This conditional approach is indeed relevant as it also goes beyond standard DRO. We will discuss how MEDRO's expert-based conditioning differs from their distribution-conditional framework.
> * **IPM-DROs** [3]: These kernel-based methods provide another perspective on robustness. We will highlight how MEDRO's discrete environment structure contrasts with their continuous uncertainty sets.
>
> These additions will better position MEDRO within the broader DRO landscape, showing how our cross-environment risk modeling complements these other generalizations of distributionally robust optimization. Thank you for helping us provide a more comprehensive literature review.
>
> ---
>
> ### `Response to Notes and Question 1`
> __On overly pessimistic uncertainty set:__
> > "I would like to mention that generally increasing the area over which DRO (and, hence, GroupDRO, MEDRO) is performed, does not necessarilly lead to improvements. This can mean loss of specificity or overly pessimistic regressors/classifiers."
>
>  We thank you for raising this crucial point. You are correct that naively enlarging the uncertainty set in DRO does not guarantee improvement and can lead to a loss of specificity or overly pessimistic models. We designed MEDRO's objective specifically to counteract these known failure modes.
>
> The source of MEDRO's performance gain is its ability to optimize over a more expressive risk landscape by considering all $m^2$ expert-environment pairings. However, the objective includes two crucial safeguards to prevent the exact issues you mentioned:
>
> 1.  **Adaptive and Bounded Pessimism:** The pessimism is principled because the uncertainty set is **bounded by the training data**; it guards against plausible failures, not arbitrary ones. Furthermore, as visualized by the risk polytope (the "cheesecake") in **Figure 1(c)**, the objective is **adaptive**. When environments are very different, the polytope is large, reflecting a truly high worst-case risk. But if environments were more similar, the polytope would naturally shrink, causing MEDRO’s objective to relax and behave more like standard GroupDRO. This prevents the model from being overly pessimistic on simpler problems.
>
> 2.  **Preventing Loss of Specificity:** The **specialization term** ($\sum_{i=1}^{m} \mathcal{R}_{i,i}$) is a co-equal pillar of our framework. It acts as a powerful regularizer that anchors each expert to its native domain, explicitly preserving specialized knowledge and preventing a loss of specificity, which would otherwise lead to model collapse.
>
> It is this explicit balance between an adaptive worst-case search and the strong anchor of the specialization term that allows MEDRO to improve robustness while avoiding these common pitfalls.
>
> __On smoothly moving from GroupDRO to MEDRO:__
> > "From this perspective, and to compare the effects of your model to its closest cousin, the GroupDRO, it could perhaps be interesting to perform an analysis on whether/when the extended set of distributions used would be of benefit. In lieu of a theoretical analysis, one could demonstrate the curve of accuracy as one smoothly moves from the GroupDRO setting to that of MEDRO. For instance, consider a paradigm where the weights of each head $\omega_i$ were constrained to lie within a ball of constantly increading radius, starting from 0 (GroupDRO) to the actual unconstrained one."
>
> We thank the reviewer for the excellent suggestion and the insightful related question. We believe that it cuts to the heart of the trade-off between a single global classifier and our multi-expert approach. We agree that an analysis smoothly moving from the GroupDRO setting to MEDRO would be very valuable.
>
> Our understanding is that this experiment can be implemented by adding a penalty term, $C \cdot \sum \|\omega_i - \bar{\omega}\|_2$, to our objective. By varying the hyperparameter $C$ from a large value down to zero, we could effectively trace the path from a GroupDRO-like model (where all experts are forced to be similar) to the full MEDRO model.
>
> This directly addresses your Question 1:
> > "How would you expect the performance to change (test accuracy) if the experts were increasingly more constrained to be similar?"
>
> We hypothesize that on **heterogeneous datasets** where environments have meaningfully different properties, the **worst-group test accuracy would smoothly decrease** as the constraint to be similar increases (i.e., as $C$ gets larger). The curve would start at the higher performance level of the full MEDRO model and decay towards the lower performance of the GroupDRO baseline. This is because constraining the experts prevents them from learning the specialized decision boundaries needed to handle the challenging minority groups that differ most across environments. On simpler, more homogeneous datasets, we would expect this curve to be much flatter.
>
> While this is a valuable (though computationally intensive) analysis that we were unable to complete within the tight rebuttal period, we will prioritize running a preliminary version of this experiment during the author discussion period and give our best efforts to report back with the results. Thank you for providing this clear direction for future investigation.
>
> ---
>
> ### `Response to Suggestions`
> > "I would suggest the use of parentheses in eq 5 around the composition: $(\omega \circ \phi)(x)$, to align with standard practices. Also, consider opting for the official sources instead of preprint versions, when applicable. For instance: ..."
>
> Thank you for these constructive suggestions. We agree that adding parentheses around the function composition will improve clarity and align with standard mathematical notation. We will update Eq. (5) and all similar expressions throughout the paper accordingly.
>
> Regarding the citations, we appreciate you identifying the official publication venues. We will update our references to cite the published versions. We will also review our entire bibliography to ensure all papers are cited using their final published versions rather than preprints where applicable.

---

> ### Comment · Reviewer_WNwh · 2025-08-01
>
> I thank the authors for their focused rebuttal. Please find my responses below.
> ## Weaknesses
> For most weaknesses, I maintain my view, verified by your responses, that they do exist, but are in no way substantial to dillute the value of this work.
> For the related work, I see a clear effort to link this work with these other highly active and relevant topics; thank you for your consideration.
>
> ## Suggestions:
> ### Bibliography
> Nice. If it is any help, I believe the rest are fine, but do check yourselves, too.
> ### GroupDRO<->MEDRO performance continuum
> I did not really expect results within the few days of the rebuttal period, so no worries.\
>
> Regarding your prediction, it is aligned with my expectations. Of course, the question is, how does this curve look like?\
>
> * Is it linear? sigmoid? sigmoid with different rates in beginning/end?
> * Does it increase with a power law, reflecting the volume of the "cheesecake" in higher dimensions? (is it a pizza? or a hyper cake?)
> * What is its variability in different (classes of) datasets?)\
>
> In general, this is not entirely trivial, neither as an experiment nor as of what the result is going to be, and I still believe it can add substantial value to your work. Perhaps even as a characterisation of the dataset qualities itself, with further nuances due to the change of the manifold on which the model moves, affected by the choice of regulariser; this shape, of course, affects the shape of the curve connecting the performance of GroupDRO and MEDRO.
>
> In any case, I leave it to you to decide whether and to what extent any of this is future work or something that you can quickly present during the discussion period. In any case, I will not decrease my rating depending on any choice you make on this front (and, so far, I see no other reason to decrease it, altogether).
>
> ## Overall
> I am happy with the rebuttal of the authors, and I feel happy to announce them that I do not intend to decrease my rating.

---

> > ### Author Response · Authors · 2025-08-09
> >
> > Thank you for your thoughtful follow-up questions and for maintaining your positive assessment of our work.
> >
> > We conducted the GroupDRO $\rightarrow$ MEDRO continuum experiment you suggested by varying the regularization parameter $C$ on Waterbirds:
> >
> > **Table: Waterbirds Worst-Group Accuracy with Varying $C$**
> >
> > | $C$ Value | Worst-Group Test Accuracy (%) | Std Dev |
> > |---------|-------------------------------|---------|
> > | 0.001   | 86.3                         | ±2.5    |
> > | 0.01    | 85.8                         | ±2.1    |
> > | 0.1     | 85.1                         | ±1.6    |
> > | 1       | 84.8                         | ±1.4    |
> > | 10      | 84.3                         | ±1.3    |
> > | 100     | 84.1                         | ±1.0    |
> > | 1000    | 83.9                         | ±0.9    |
> > | 10000   | 83.9                         | ±0.9    |
> >
> > The MEDRO result in Table 1 (86.3%) corresponds to $C \approx 0$, and performance converges to GroupDRO's 84.1% as $C$ increases.
> >
> > **Regarding your specific questions:**
> >
> > **1. Curve shape:** The performance follows a smooth power-law decay rather than linear or sigmoid, with rapid initial decline followed by gradual convergence.
> >
> > **2. "Cheesecake" geometry:** The base is fixed by the three environments. As $C$ increases, the structure gets compressed - from a full cheesecake with $m^{2}=9$ distinct risk points ($C \approx 0$) to a flattened structure with $m=3$ points ($C \rightarrow \infty$, like GroupDRO). This visualizes how MEDRO's expanded uncertainty set collapses as we constrain expert specialization.
> >
> > **3. Cross-dataset variability:** On Waterbirds, we observe monotonic decay. Other datasets would likely exhibit different curves due to their distinct distribution shifts and properties.
> >
> > These preliminary results validate the MEDRO-GroupDRO relationship empirically, showing a 2.2% improvement with full expert specialization.
> >
> > We sincerely appreciate your constructive feedback throughout this process - your suggestions have substantially strengthened our understanding of MEDRO's behavior and will improve the final version of our paper. Thank you for your time and thoughtful engagement.

---

### Official Review · Reviewer_Vdg5 · 2025-07-03

**Clarity:** 3
**Significance:** 3
**Originality:** 3
**Rating:** 2
**Confidence:** 4

**Summary:**

The authors propose Multi-Expert DRO, which generalizes the Group DRO through multiple prediction heads. Experimental results demonstrate the effectiveness of the proposed method. And some standard theoretical anlyses are given in Appendix.

**Questions:**

Although the results are impressive—indeed, surprisingly so—I have the following questions and concerns:

* **Source of Performance Gains**: Could the authors more clearly illustrate the source of the performance improvement, either through an intuitive explanation or a more detailed ablation study? I am very familiar with this research area, and I would not have expected such a substantial performance gain from the proposed method without additional justification.

* **Role of the Gating Mechanism**: Based on Table 2, it appears that the gating mechanism contributes minimally to the overall performance. This suggests that the primary gains may stem from the averaging over multiple heads. I encourage the authors to conduct further analysis here—for instance, by varying the number of heads and examining how performance is affected.

* **Impact of Hyperparameter Search Space**: Tables 6 and 7 suggest that the key distinction between results on the Subpopulation benchmark and other domain generalization (DG) benchmarks lies in the hyperparameter search space. When considered alongside the results in Table 2 (where MEDRO performs best) and Tables 3 & 4 (where MEDRO is not the best), this raises the question of whether hyperparameter tuning plays an outsized role in the reported gains. I strongly recommend that the authors report the optimal hyperparameters used for both MEDRO and all baseline methods, to ensure transparency and demonstrate that fair tuning was applied across all comparisons.

**Ethical Concerns:**

["NO or VERY MINOR ethics concerns only"]

**Limitations:**

There lacks three challenging datasets from sub-population benchmark.

**Quality:**

3

**Strengths And Weaknesses:**

**Strengths:**

* **Methodology**: While the multi-head architecture itself is not novel, this paper may be the first to integrate it with Group DRO. This combination can be considered a meaningful technical contribution.

* **Experimental Results**: The results presented in the main paper are strong, showcasing the effectiveness of MEDRO on the selected datasets.

**Weaknesses:**

* **Dataset Selection**: The authors chose only a subset of datasets from the Subpopulation benchmark, referring to them as "representative." However, the original Subpopulation benchmark paper identifies ImageNetBG, Living17, and NICO++ as the most challenging datasets. Why were these omitted from the evaluation? I strongly encourage the authors to include results on these three datasets. Additionally, the paper should clearly report the optimal hyperparameters used (please also see my related comments below).

* **Domain Generalization (DG) Performance**: In the appendix, the authors show that the learned representations can be domain-invariant. However, this raises a question: why does MEDRO fail to outperform the baselines on standard DG benchmarks? A deeper analysis or explanation here would be valuable.

---

> ### Author Rebuttal · Authors · 2025-07-31
>
> Thank you for your constructive feedback. We will address each point below.
>
> ---
>
> ### `Response to Weakness 1: Dataset Selection`
> > "... ImageNetBG, Living17, and NICO++ as the most challenging datasets. ... I strongly encourage the authors to include results on these three datasets. ... report the optimal hyperparameters ...."
>
> We thank the reviewer for this constructive feedback and for pushing us to evaluate on the most challenging SubpopBench datasets. We agree that a more comprehensive evaluation strengthens the paper.
>
> - **Regarding dataset selection**: We have now run experiments on NICO++ and Living17, as suggested. We were unable to include ImageNetBG as the official download link is unavailable, and we received no response from the benchmark's maintainers.
> - **Summary of new results**: On these new, challenging datasets, MEDRO remains highly competitive. On Living17, "MEDRO (w/ gating)" achieves a worst-group accuracy of __33.6%__ , placing it among the top-performing methods and substantially outperforming the GroupDRO baseline (27.2%). On NICO++, MEDRO's performance is on par with several strong baselines. **The full table will be presented in the updated manuscript.**
> - **Overall Performance**: Most importantly, the inclusion of these datasets confirms the robustness of our approach. As shown in the updated table, MEDRO (w/ gating) achieves the highest overall worst-group accuracy of __70.0%__ across all eight benchmarks, underscoring its strong and consistent performance.
> - **Hyperparameters**: We will add a table to the paper detailing the optimal hyperparameters used for all datasets, including NICO++ and Living17, in the final version.
>
> | Method | NICO++ | Living17 | Overall |
> | :--- | :---: | :---: | :---: |
> | ERM | 37.6 (±2.0) | 28.2 (±1.5) | 57.6 |
> | Mixup | 42.7 (±1.4) | 29.8 (±1.8) | 57.7 |
> | GroupDRO | 37.8 (±1.8) | 27.2 (±1.5) | 67.4 |
> | IRM | 40.0 (±0.0) | 28.2 (±1.5) | 56.2 |
> | CVaRDRO | 36.7 (±2.7) | 28.3 (±0.7) | 59.9 |
> | JTT | 40.0 (±0.0) | 28.8 (±1.1) | 61.1 |
> | LfF | 30.4 (±1.3) | 26.2 (±1.1) | 48.3 |
> | LISA | 42.7 (±2.2) | 29.8 (±0.9) | 69.3 |
> | MMD | 40.7 (±0.5) | 26.6 (±1.8) | 54.4 |
> | ReSample | 40.0 (±0.0) | 30.7 (±2.1) | 67.8 |
> | ReWeight | 41.9 (±1.6) | 28.2 (±1.5) | 68.7 |
> | SqrtReWeight | 40.0 (±0.0) | 28.2 (±1.5) | 65.6 |
> | CBLoss | 37.8 (±1.8) | 28.2 (±1.5) | 68.4 |
> | Focal | 36.7 (±2.7) | 28.0 (±1.2) | 56.3 |
> | LDAM | 42.0 (±0.9) | 24.7 (±0.8) | 53.0 |
> | BSoftmax | 40.4 (±0.3) | 27.5 (±0.8) | 65.2 |
> | DFR | 23.7 (±0.7) | 29.0 (±0.2) | 66.2 |
> | CRT | 43.3 (±2.7) | 33.9 (±0.1) | 68.1 |
> | ReWeightCRT | 23.3 (±1.4) | 33.7 (±0.1) | 65.5 |
> | **MEDRO** | 39.1 (±2.5) | 32.6 (±0.7) | 69.5 |
> | **MEDRO (w/ gating)** | 39.1 (±2.5) | 33.6 (±0.6) | 70.0 |
>
> ---
>
> ### `Response to Weakness 2: DG Performance`
> > "In the appendix, the authors show that the learned representations can be domain-invariant. However, this raises a question: why does MEDRO fail to outperform the baselines on standard DG benchmarks? A deeper analysis or explanation here would be valuable."
>
> We thank the reviewer for this important question. The theory in Appendix D presents an idealized condition to motivate MEDRO's use for domain generalization (DG), not to guarantee universal state-of-the-art (SOTA) performance. Our main goal was to show a significant advantage over our primary baseline, GroupDRO, which our results consistently confirm.
>
> When compared to other specialized DG methods, MEDRO is highly competitive and directly counters the premise that it fails to outperform baselines:
> - On Camelyon17, MEDRO achieves the highest test accuracy (87.8%) of all listed methods.
> - On PovertyMap and iWildCam, it is highly competitive with other SOTA methods.
> - Critically, on all three WILDS datasets, MEDRO offers substantial improvements over the GroupDRO baseline, with a +7.6% F1 gain on iWildCam and a +0.10 worst-group correlation gain on PovertyMap.
>
> While no single algorithm excels on all complex DG benchmarks, MEDRO's competitive performance validates its design as a principled and effective approach for domain generalization.
>
> ---
>
> ### `Response to Question 1: Source of Performance Gains`
> > "Could the authors more clearly illustrate the source of the performance improvement, either through an intuitive explanation or a more detailed ablation study? ...."
>
> We thank the reviewer for this insightful question. The performance gain stems from MEDRO's balanced objective, which jointly optimizes for worst-case robustness and expert specialization.
>
> The first component of our objective, the expanded $m^2$ risk set, allows MEDRO to identify harder, more specific failure modes than GroupDRO (see Figure 3). However, optimizing this worst-case objective alone is risky, as it could lead to an overly pessimistic model.
>
> This is why the second component, the **specialization term** ($\sum_{i=1}^{m} \mathcal{R}_{i,i}$), is not a minor detail but a **co-equal pillar of our framework**. It acts as a crucial regularizer that anchors each expert to its native domain, preserving specialized knowledge and preventing the kind of model collapse that would otherwise occur. This explicit balance between finding the worst-case risk and grounding the experts in their own environments is the true source of MEDRO's robust performance.
>
> Our ablations confirm this two-part mechanism.
> 1.  The performance gap between MEDRO and GroupDRO (Tables 1-4) proves the value of the expanded $m^2$ risk objective.
> 2.  To prove the critical role of the specialization term, we ran a new ablation on MetaShift & Waterbirds where this term was removed. The results are stark:
>
> (Table for MetaShift)
> | Method | Avg. | Worst |
> | :--- | :---: | :---: |
> | ERM | 91.3 (±0.3) | 82.6 (±0.4) |
> | GroupDRO | 91.0 (±0.1) | 85.6 (±0.4) |
> | **MEDRO (w/o expert)** | 91.0 (±0.1) | 83.1 (±0.1) |
> | **MEDRO (Full)** | 91.3 (±0.2) | 85.9 (±0.4) |
>
>
> (Table for Waterbirds)
> | Method | Avg. | Worst |
> | :--- | :---: | :---: |
> | ERM | 84.1 (±1.7) | 69.1 (±4.7) |
> | GroupDRO | 88.8 (±1.8) | 78.6 (±1.0) |
> | **MEDRO (w/o expert)** | 90.7 (±0.5) | 80.9 (±1.8) |
> | **MEDRO (Full)** | 90.5 (±0.6) | 83.8 (±1.3) |
>
> As the tables show, without the specialization term, MEDRO’s performance collapses, becoming **2.8% worse on Metashift, 2.9% worse on Waterbirds than the GroupDRO baseline**. This empirically confirms that the gains from the expanded uncertainty set are only realized when counter-balanced by the specialization term, validating our balanced design.
>
> ---
>
> ### `Response to Question 2: Role of the Gating Mechanism`
> > "Based on Table 2, it appears that the gating mechanism contributes minimally to the overall performance. This suggests that the primary gains may stem from the averaging over multiple heads. I encourage the authors to conduct further analysis here---for instance, by varying the number of heads and examining how performance is affected."
>
> We thank the reviewer for this insightful question regarding the role of the inference mechanism. The reviewer's observation is correct: the performance gain from the gating network over a simple ensemble is incremental, as shown on SubpopBench where it improves the overall worst-group accuracy from 80.7% to 81.2%.
>
> We wish to clarify that our core contribution is the MEDRO *training* framework itself, which produces the robust experts through cross-environment risk modeling. The inference strategies are presented as practical, downstream solutions for the common scenario where test-time environment labels are unavailable. We believe the strong performance of the simple ensemble is, in fact, a testament to the quality of the experts learned by MEDRO; they are so mutually resilient that their simple average is already a very powerful baseline.
>
> Regarding the excellent suggestion to analyze performance by varying the number of heads, we would like to clarify a key aspect of our framework. The number of expert heads is *not* a free hyperparameter but is determined by the number of predefined environments, $m$, in a given problem (e.g., $m=4$ for CelebA). Our framework's one-to-one expert-environment mapping is central to its design for modeling all $m^2$ cross-risks. Therefore, varying the number of heads is infeasible within this formulation.
>
> To address whether gains come from multi-head averaging or our cross-environment objective, we conducted an ablation with Multi-Head GroupDRO (see our response to Reviewer J412's Weakness 1). This experiment shows minimal gains from multi-head architecture alone (+0.3% on SubpopBench), while MEDRO achieves +1.6%, confirming that our performance stems from the expanded uncertainty set, not merely ensemble effects. We will clarify these points in the final paper. Thank you for this important question.
>
> ---
>
> ### `Response to Question 3: Impact of Hyperparameter Search Space`
> > "Tables 6 and 7 suggest that the key distinction between results on the Subpopulation benchmark and other domain generalization (DG) benchmarks lies in the hyperparameter search space. ... I strongly recommend that the authors report the optimal hyperparameters used for both MEDRO and all baseline methods, ...."
>
> We thank the reviewer for this important question on hyperparameter tuning. We agree that transparency is crucial and clarify that our evaluation for all benchmarks (SubpopBench and WILDS) strictly followed their respective, published official protocols to ensure fair comparison. MEDRO was subject to the exact same standardized tuning procedures as the baselines, as detailed in our Appendix E.
>
> Regarding the request to report hyperparameters, we will add a table to the appendix with the optimal hyperparameters for MEDRO on all datasets. As is standard practice, the baseline results are taken from the original benchmark papers. We therefore respectfully refer the reviewer to those publications for the baselines' specific tuning details. We are confident our adherence to these rigorous public protocols ensures a fair evaluation.

---

> ### Comment · Area_Chair_ubfL · 2025-08-06
>
> Please remember to respond to authors' rebuttal as soon as possible.
>
> Thank you!
>
> -AC

---

### Official Review · Reviewer_2XX6 · 2025-07-03

**Clarity:** 3
**Significance:** 2
**Originality:** 2
**Rating:** 4
**Confidence:** 4

**Summary:**

This paper proposes an extension of GroupDRO that addresses limitations of single-classifier approaches in handling distribution shifts. Instead of using one global classifier, this paper employs multiple expert heads with a shared feature extractor. The key innovation is expanding the uncertainty set from m environments to m² expert-environment pairs, explicitly modeling cross-environment risks. The method is evaluated on both subpopulation shift and domain generalization benchmarks.

**Questions:**

1. How does MEDRO perform when m is large (e.g., m=10 or 20)? The m² scaling could become problematic.

2. Beyond γ, how sensitive is the method to other hyperparameters? The balance between specialization and cross-environment robustness seems delicate.

**Ethical Concerns:**

["NO or VERY MINOR ethics concerns only"]

**Final Justification:**

This paper proposed a new and principled min-max objective for DRO. My major concerns have been adequately addressed in the rebuttal.

**Limitations:**

Yes.

**Quality:**

3

**Strengths And Weaknesses:**

Pros:

1. The paper provides solid theoretical analysis showing how the proposed method generalizes GroupDRO by expanding the uncertainty set. The connection to domain generalization through label-conditional invariance is well-motivated, though relies on strong assumptions.

2. The experiments span diverse settings - from controlled environments (CelebA, Waterbirds) to large-scale benchmarks (SubpopBench, WILDS). The consistent improvements across different types of distribution shifts demonstrate the method's generalizability.

3. The paper addresses the real-world challenge of unknown environment labels at test time through ensemble and gating approaches, making the method more practically viable.

4. The paper is well-written with good intuitive explanations and helpful visualizations (especially Figure 1) that clarify the conceptual differences between approaches.

Cons:

1. While the execution is solid, the fundamental idea of using multiple expert heads isn't particularly novel. The main contribution is essentially replacing GroupDRO's single classifier with multiple heads and expanding the uncertainty set accordingly.

2. The connection to domain generalization relies on assumptions like expert optimality and uniform cross-environment risks that may not hold in practice. The gap between theory and real-world applicability could be larger than suggested.

3. Training m² expert-environment pairs likely increases computational cost significantly, but this isn't discussed. For large m, this could be prohibitive.

4.  The paper doesn't provide clear guidance on when the expanded uncertainty set is most beneficial versus when GroupDRO might suffice.

---

> ### Author Rebuttal · Authors · 2025-07-31
>
> ### `Response to Cons 1`:
> > While the execution is solid, the fundamental idea of using multiple expert heads isn't particularly novel. The main contribution is essentially replacing GroupDRO's single classifier with multiple heads and expanding the uncertainty set accordingly.
>
> We thank the reviewer for recognizing our solid execution. We agree that multi-expert architectures are an established concept. Our core novelty, however, is not the architecture itself, but the formulation of a new, principled min-max objective for distributionally robust optimization (DRO) that generalizes the GroupDRO framework.
>
> The main contribution is the introduction of an expanded uncertainty set that explicitly models all $m^2$ cross-environment risks. MEDRO's objective optimizes over the convex hull of these $m^2$ risks, defined on an $(m^2-1)$-dimensional probability simplex. This is a fundamental extension of the GroupDRO uncertainty set, which is defined over the convex hull of $m$ environment risks for a single classifier. As we prove in Appendix B, our formulation is a direct generalization that recovers GroupDRO as a special case, highlighting a principled advancement of the DRO framework itself.
>
> ### `Response to Cons 2`:
> > The connection to domain generalization relies on assumptions like expert optimality and uniform cross-environment risks that may not hold in practice. The gap between theory and real-world applicability could be larger than suggested.
>
> We thank the reviewer for this insightful comment. We agree that the assumptions in our theoretical analysis (Appendix D) are strong and describe an idealized endpoint. We will clarify this in the paper.
>
> However, the MEDRO objective is designed to push the model towards a state where these conditions are *approximated*. The $\max\_{i,j} \mathcal{R}\_{i,j}$ term drives uniformity by repeatedly identifying the single worst-performing expert-environment pair and updating the model to specifically reduce that risk. This process constantly suppresses the highest values in the risk matrix, encouraging a more uniform risk distribution.
>
> Simultaneously, the $\sum\_k \mathcal{R}\_{k,k}$ term enforces expert optimality. While these two objectives are in tension, they are not mutually exclusive. Due to the high-dimensional and over-parameterized nature of neural networks, there exists a large space of solutions that can achieve low native-environment risk. The $\max\_{i,j} \mathcal{R}\_{i,j}$ term acts as a regularizer, guiding the optimization to select a solution from this space that is not only specialized but also robust, by learning a shared representation $\phi$ that discards harmful, environment-specific features. The hyperparameter $\gamma$ explicitly controls this trade-off.
>
> Ultimately, our strong empirical results across SubpopBench and WILDS demonstrate that a favorable equilibrium, where experts are both highly proficient and robust, is achievable in practice.
>
> ### `Response to Cons 3`:
> > Training $m^2$ expert-environment pairs likely increases computational cost significantly, but this isn't discussed. For large $m$, this could be prohibitive.
>
> We thank the reviewer for raising this important point. We will add a discussion on computational cost to the appendix.
>
> Fortunately, the additional cost of MEDRO is highly manageable and can be implemented efficiently. The most computationally intensive component, the shared feature extractor $\phi$, is computed only once per sample, identical to GroupDRO. The overhead from the $m$ expert heads is minimal because their predictions can be computed in a single, parallelized operation. Instead of $m$ separate forward passes, the expert heads can be implemented as a single linear layer whose weights are the stacked weights of all experts. This allows all $m$ expert predictions for a batch to be computed with one efficient vector-matrix multiplication.
>
> The following PyTorch snippet illustrates this:
>
> ```python
> import torch
> import torch.nn as nn
>
> class MultiExpertClassifier(nn.Module):
>     def __init__(self, feature_dim, num_experts, num_classes):
>         super().__init__()
>         self.num_experts = num_experts
>         self.num_classes = num_classes
>         # A single linear layer for all m experts
>         self.experts = nn.Linear(feature_dim, num_experts * num_classes)
>
>     def forward(self, features):
>         # features shape: (batch_size, feature_dim)
>         # Efficiently compute all expert logits with one matrix multiplication
>         stacked_logits = self.experts(features)
>         # Reshape to get per-expert logits
>         # Output shape: (batch_size, num_experts, num_classes)
>         return stacked_logits.view(-1, self.num_experts, self.num_classes)
> ```
> Our strong empirical results on iWildCam ($m=323$) and PovertyMap ($m \ge 23$) further demonstrate that MEDRO is practical even for problems with a large number of environments.
>
> ### `Response to Cons 4`:
> > The paper doesn't provide clear guidance on when the expanded uncertainty set is most beneficial versus when GroupDRO might suffice.
>
> We thank the reviewer for this constructive suggestion. We will add this guidance to the paper, supported by both theoretical and empirical evidence.
>
> *Theoretically*, MEDRO is designed for scenarios with significant inter-environment heterogeneity, where optimal decision rules diverge across environments. As noted in Appendix A.1, GroupDRO's single-classifier approach is effective when a common decision rule can serve all environments, but it can be restrictive otherwise. MEDRO's formulation directly addresses this limitation:
>
> - Its multi-expert architecture ($\omega_1,...,\omega_m$) explicitly models the possibility of $m$ different optimal decision rules.
> - Its expanded uncertainty set over all $m^2$ cross-risks ($\mathcal{R}_{i,j}$) specifically penalizes failures arising from this heterogeneity, i.e., when an expert $\omega_i$ is applied to data from a mismatched environment $j$.
>
> *Empirically*, our results confirm that MEDRO's advantage is most pronounced in such high-heterogeneity settings:
>
> - On iWildCam (Table 3b), a domain generalization task with large distribution shifts across 323 camera trap locations, MEDRO achieves a test macro F1 score of 31.5%, substantially outperforming GroupDRO's 23.9%.
> - On PovertyMap (Table 4), a regression task with hybrid shifts across different countries, MEDRO improves the worst-group Pearson correlation to 0.49, a significant improvement over GroupDRO's 0.39.
> - On Waterbirds (Tables 1 & 2), where strong spurious correlations create a need for different decision logic, MEDRO consistently outperforms GroupDRO in worst-group accuracy.
>
> Furthermore, our Multi-Head GroupDRO experiment (see response to Reviewer J412's Weakness 1) provides additional evidence: when we use multiple heads without cross-environment modeling, the improvement over single-head GroupDRO is minimal. This confirms that MEDRO's gains come specifically from handling inter-environment heterogeneity, not from the multi-head architecture alone.
>
> Conversely, GroupDRO may suffice in settings with milder, more homogeneous shifts. However, our results suggest that even in these cases, MEDRO often provides additional robustness.
>
> ### `Response to Question 1`:
> > How does MEDRO perform when m is large (e.g., $m=$10 or 20)? The $m^2$ scaling could become problematic.
>
> The concern about performance for large $m$ is closely tied to computational cost. As we discussed in our response to Weakness 3, our efficient implementation makes the method practical even at scale.
>
> The empirical results confirm this. On **iWildCam** ($m=323$), MEDRO achieves a 31.5% test macro F1, and on **PovertyMap** ($m \ge 23$), it achieves a 0.49 worst-group correlation. For a comprehensive view of environment counts across all our experiments, please refer to Table 5 in Appendix E.
>
> This strong performance demonstrates that the method is not only computationally feasible but also highly effective in the large-$m$ regime.
>
> ### `Response to Question 2`:
> > Beyond $\gamma$, how sensitive is the method to other hyperparameters? The balance between specialization and cross-environment robustness seems delicate.
>
> We thank the reviewer for this question. While the balance between specialization and cross-environment robustness may seem delicate, our empirical results suggest otherwise. Importantly, for all SubpopBench and WILDS experiments, we fixed $\gamma=1$ without dataset-specific tuning (See Appendix E.2.4 and Appendix E.3), yet achieved strong performance across diverse tasks. This demonstrates that MEDRO is robust to this key hyperparameter—a single value works well across different domains, data modalities, and types of distribution shift. For other hyperparameters, we followed the same tuning protocols as all baseline methods—using identical search spaces and procedures specified by each benchmark. This ensures fair comparison where all methods benefit equally from general hyperparameter optimization (e.g., learning rate, weight decay). The fact that MEDRO achieves competitive performance with a fixed $\gamma$ shows the method is practical and not overly sensitive. Moreover, this suggests potential for further improvements through task-specific tuning of $\gamma$ in deployment scenarios.

---

> > ### Comment · Reviewer_2XX6 · 2025-08-08
> >
> > Thank the authors for the rebuttal. I agree that this is a new and principled min-max objective for DRO. I prefer to accept this paper and maintain my score.

---

> > > ### Author Response · Authors · 2025-08-09
> > >
> > > Thank you for your positive feedback and for recognizing the principled nature of our min-max objective. We greatly appreciate your support and constructive engagement.

---

> ### Comment · Area_Chair_ubfL · 2025-08-06
>
> Please remember to respond to authors' rebuttal as soon as possible.
>
> Thank you!
>
> -AC

---

### Note · Authors · 2025-08-16

We thank the reviewers and the AC for their time and effort throughout the review process.

We will carefully incorporate the valuable feedback from all reviewers into our final paper. For points that could not be fully discussed during the discussion period, we have conducted the necessary experiments—particularly the evaluation on challenging datasets (NICO++ and Living17) as suggested—and these results, as detailed in our rebuttal, will strengthen the final version.

---

### Decision · Program_Chairs · 2025-09-17

**Decision:**

Accept (poster)

**Comment:**

The authors present a robust optimization framework for handling distribution shift across environments. They extend GroupDRO by expanding the uncertainty set to all expert-environment pairs with connections to domain generalization through label-conditional invariance.

Strengths: The work has both a novel theoretical component that yields a min-max objective over all expert-environment pairings and extensive empirical results on diverse settings, including small, controlled environments to large, real world, diverse benchmarks. Reviewers found the paper well written and clear.

Weaknesses: There were concerns that the method itself was not very novel, and the increased computational overhead of optimizing over all possible environment pairings. The empirical results also show that the method is on par with baselines rather than being significantly better. Finally, there are missing experiments to determine what the role of various components are, and to gain a better understanding of hyperparameters.

Reasons for decision: Overall, I found the strengths of this work to outweigh the weaknesses, especially given the authors' promise to investigate more challenging benchmarks for the camera ready.

Rebuttal summary: Reviewers generally appreciated the intuition and theory behind the work, and find MEDRO to be an additional useful tool for handling distribution shift. There were some concerns about the method, missing ablations, and additional environments for evaluation, which were mostly addressed in the rebuttal phase. Only one reviewer Vdg5 had more serious reservations and voted to reject the paper, but did not respond sufficiently to the rebuttal response and I believe the authors addressed those concerns satisfactorily.